# DP$^2$O-SR: Direct Perceptual Preference Optimization for Real-World Image Super-Resolution

**Rongyuan Wu**[1,2], **Lingchen Sun**[1,2], **Zhengqiang Zhang**[1,2], **Shihao Wang**[1],
**Tianhe Wu**[2,3], **Qiaosi Yi**[1,2], **Shuai Li**[1], **Lei Zhang**[1,2,†]

[1]The Hong Kong Polytechnic University      [2]OPPO Research Institute
[3]City University of Hong Kong
[†]Corresponding author
https://github.com/cswry/DP2O-SR

## Abstract

Benefiting from pre-trained text-to-image (T2I) diffusion models, real-world image super-resolution (Real-ISR) methods can synthesize rich and realistic details. However, due to the inherent stochasticity of T2I models, different noise inputs often lead to outputs with varying perceptual quality. Although this randomness is sometimes seen as a limitation, it also introduces a wider perceptual quality range, which can be exploited to improve Real-ISR performance. To this end, we introduce Direct Perceptual Preference Optimization for Real-ISR (DP$^2$O-SR), a framework that aligns generative models with perceptual preferences without requiring costly human annotations. We construct a hybrid reward signal by combining full-reference and no-reference image quality assessment (IQA) models trained on large-scale human preference datasets. This reward encourages both structural fidelity and natural appearance. To better utilize perceptual diversity, we move beyond the standard best-vs-worst selection and construct multiple preference pairs from outputs of the same model. Our analysis reveals that the optimal selection ratio depends on model capacity: smaller models benefit from broader coverage, while larger models respond better to stronger contrast in supervision. Furthermore, we propose hierarchical preference optimization, which adaptively weights training pairs based on intra-group reward gaps and inter-group diversity, enabling more efficient and stable learning. Extensive experiments across both diffusion- and flow-based T2I backbones demonstrate that DP$^2$O-SR significantly improves perceptual quality and generalizes well to real-world benchmarks.

## 1 Introduction

Image super-resolution (ISR) [13, 60, 25, 59, 10, 23, 56, 16] aims to reconstruct high-resolution (HR) images from low-resolution (LR) inputs. Traditional methods emphasize pixel-level accuracy but often produce over-smoothed results that lack realistic textures. To address this, recent approaches [22, 44, 43, 55, 24, 8] have shifted toward improving perceptual quality, which is particularly important for real-world ISR (Real-ISR) tasks where degradations are complex and typically unknown. Generative models [35, 17], especially large-scale pre-trained text-to-image (T2I) diffusion models such as Stable Diffusion (SD) [1] and FLUX [20], have demonstrated strong potential for Real-ISR [42, 52, 28, 48, 37, 33, 2] due to their capacity to synthesize plausible and diverse details. However, these models are inherently stochastic: different noise inputs can lead to significantly different output qualities. While this randomness is often considered as a drawback [37], it also introduces a broader perceptual quality range, which can be viewed as a source of supervision, enabling preference-driven optimization to better exploit T2I model's generative capability.

39th Conference on Neural Information Processing Systems (NeurIPS 2025).

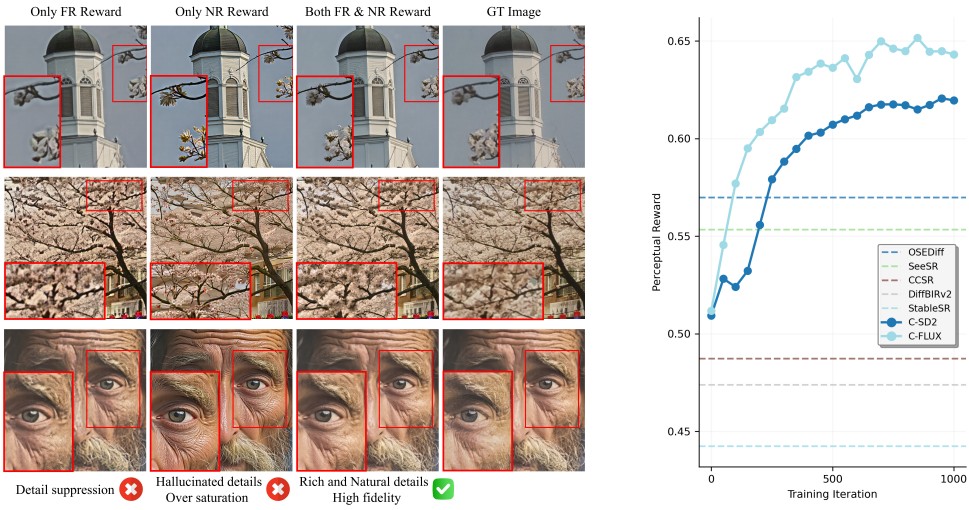

<table>
<tr><td>Only FR Reward</td><td>Only NR Reward</td><td>Both FR & NR Reward</td><td>GT Image</td></tr>
</table>

Detail suppression ❌    Hallucinated details ❌    Rich and Natural details ✅
Over saturation          High fidelity

(a) Effect of Perceptual Reward Types.

(b) DP²O-SR Generalization Performance.

Figure 1: (a) Visual results of models trained with FR, NR, and hybrid rewards. FR reward suppresses detail, NR reward encourages hallucinations, while the hybrid reward preserves structure and improves realism. (b) DP²O-SR significantly boosts perceptual quality on the out-of-domain RealSR benchmark [5], improving both small and large generative Real-ISR models after only 500 steps. Larger models like C-FLUX benefit more from preference supervision.

To harness this diversity, we propose **DP²O-SR**—**D**irect **P**erceptual **P**reference **O**ptimization for Real-I**SR**—a training framework that aligns generative ISR models with human-like perceptual preferences. Instead of relying on costly human annotations, we construct a perceptual reward using image quality assessment (IQA) models trained on large-scale human preference data. This reward integrates both full-reference (FR) and no-reference (NR) metrics. FR metrics promote structural fidelity and help suppress hallucinated content, whereas NR metrics encourage realism and aesthetic coherence. By combining FR and NR IQA metrics, the hybrid reward provides a balanced signal that supports both accuracy and naturalness. As illustrated in Fig. 1(a), models trained with only FR metrics tend to produce oversmoothed outputs, while those trained with NR metrics alone may generate hallucinated details. In contrast, the hybrid reward leads to outputs with rich and natural details while remaining structurally consistent with ground-truth (GT).

Unlike Diff-DPO [40], which constructs a single best-vs-worst pair from outputs of different models, we sample multiple outputs from a single model using different noise seeds. These outputs are ranked by perceptual reward, and preference pairs are formed by sampling from the top-$N$ and bottom-$N$ candidates. This richer supervision captures finer perceptual distinctions and better utilizes the diversity inherent in stochastic generation.

We systematically investigate how the number of rollouts ($M$) and the selection ratio ($N/M$) influence learning across two representative backbones: a relatively samller diffusion model (ControlNet-SD2, denoted as C-SD2) and a larger flow-based model (ControlNet-FLUX, denoted as C-FLUX). Increasing $M$ improves perceptual diversity and training stability, though with diminishing returns. The optimal $N/M$ varies by model capacity: smaller models prefer broader coverage (*e.g.*, $1/4$) for smoother gradients, while larger models benefit more from stronger contrast (*e.g.*, $1/16$), as their greater capacity enables them to learn more effectively from larger preference differences. These findings underscore the need to tailor data curation according to model scale.

Even with well-chosen $M$ and $N/M$, not all comparisons are equally informative—some are ambiguous or redundant. This motivates a more selective approach to learning. We propose *Hierarchical Preference Optimization* (HPO), which adaptively weights training pairs at two levels: intra-group, by emphasizing comparisons with larger reward gaps; and inter-group, by prioritizing inputs with greater perceptual spread. By focusing on the most informative signals, HPO improves both training efficiency and perceptual alignment.

We evaluate DP²O-SR on the out-of-domain RealSR benchmark [5], which contains real-world LR-HR pairs captured under varying focal lengths, differing from the synthetic degradations used

during training. As shown in Fig. 1(b), both C-SD2 and C-FLUX achieve significant perceptual reward improvements within the first 500 training iterations of DP²O-SR, surpassing strong baselines such as SeeSR [48] and OSEDiff [47]. Specifically, C-FLUX improves from approximately 0.51 to 0.65, while C-SD2 rises to 0.62, achieving top-2 performance early in training.

## 2 Related Work

**Real-World Image Super-Resolution.** Early deep learning-based ISR methods [13, 27, 60, 12] primarily optimize pixel-level accuracy based on simple degradation assumptions, such as bicubic downsampling. As a result, they often produce over-smoothed outputs in real-world scenarios. To enhance perceptual quality, researchers have introduced GAN-based techniques [22], including BSRGAN [55] and Real-ESRGAN [43], which use complex degradation models to better simulate complex real-world degradations. However, these GAN-based approaches often suffer from unstable training and visual artifacts [43, 24]. Recently, diffusion models [17, 35], especially large-scale pre-trained T2I models [1, 20], have achieved strong results in Real-ISR tasks [42, 37, 48, 50]. These models excel at generating realistic details, but their outputs vary across different runs for the same LR input due to the inherent sampling randomness. Many existing works treat this randomness as a limitation. Some methods aim to stabilize the generation process [37], while others train one-step models [47, 36, 6, 53, 14] to reduce stochasticity. In contrast, we view this stochasticity as a useful source of high-quality supervision and consequently develop a preference-based optimization framework to improve Real-ISR performance.

**Preference Alignment.** Aligning generative models with human preferences has become a key area of research, especially in the training of large language models (LLMs) using reinforcement learning with human feedback (RLHF) [11, 61, 32]. This process typically requires training a separate reward model, which adds significant computational cost [21]. DPO [34] provides a simpler alternative by directly optimizing the policy using preference data, without requiring an explicit reward model. Adapting these alignment techniques to diffusion models introduces additional challenges, mainly due to their iterative denoising process [29]. Recent work has extended DPO to diffusion models. For example, Diff-DPO [40] applies preference optimization across diffusion timesteps to improve visual aesthetics and prompt fidelity. Several follow-up works propose refinements, such as step-aware preference modeling [26, 18] and sample weighting based on score distributions [30]. Although DPO-based methods can be applied to Real-ISR, there lack a well-designed reward and carefully designed training strategies. We address these challenges by introducing a perceptual reward tailored to Real-ISR and by systematically exploring preference pair construction strategies beyond the conventional best-vs-worst selection. This enables more reliable learning across models with different capacities and under limited sampling budgets.

## 3 Background

**Diffusion and Flow-based Models.** Both diffusion and flow-based generative models define a stochastic interpolation between data $x^* \sim p(x)$ and noise $\epsilon \sim \mathcal{N}(0, I)$ using the unified formulation $x_t = \alpha_t x^* + \sigma_t \epsilon$ [31], where $t$ is the timestep, and $\alpha_t, \sigma_t$ are scalar scheduling coefficients. The goal is to learn a model that reverses this process to sample from the data distribution. Diffusion models [17] specify this path implicitly via a forward SDE and typically learn the score function $s_\theta(x, t) = \nabla_x \log p_t(x)$, with a standard constraint $\alpha_t^2 + \sigma_t^2 = 1$. In contrast, flow-based models [15] directly define the interpolation with $\alpha_t + \sigma_t = 1$, enabling exact transport in finite time, and learn a velocity field $v_\theta(x, t) = \frac{d}{dt} x_t$. Different scheduling schemes lead to distinct sampling trajectories.

**From RLHF to Diff-DPO.** Aligning generative models with human preferences traditionally relies on RLHF, which involves training a reward model followed by policy optimization. However, this multi-stage process is complex and unstable. DPO [34] offers a simpler alternative by directly optimizing the policy using preference data. Under the Bradley-Terry preference model [4], DPO derives a loss that encourages higher likelihood ratios for preferred samples. As derived in the work [40], the training objective of Diff-DPO is formulated as follows:

$$L_{DPO} = - \mathbb{E}_{(\boldsymbol{x}_0^w, \boldsymbol{x}_0^l) \sim D, t \sim \mathcal{U}(0, T), \boldsymbol{x}_t^w \sim q(\boldsymbol{x}_t^w | \boldsymbol{x}_0^w), \boldsymbol{x}_t^l \sim q(\boldsymbol{x}_t^l | \boldsymbol{x}_0^l)} \log \sigma(-\beta($$
$$\|\boldsymbol{\epsilon}^w - \boldsymbol{\epsilon}_\theta(\boldsymbol{x}_t^w, t)\|_2^2 - \|\boldsymbol{\epsilon}^w - \boldsymbol{\epsilon}_{\text{ref}}(\boldsymbol{x}_t^w, t)\|_2^2 - (\|\boldsymbol{\epsilon}^l - \boldsymbol{\epsilon}_\theta(\boldsymbol{x}_t^l, t)\|_2^2 - \|\boldsymbol{\epsilon}^l - \boldsymbol{\epsilon}_{\text{ref}}(\boldsymbol{x}_t^l, t)\|_2^2))), \quad (1)$$

where $\mathbf{x}_t^w = \alpha_t \mathbf{x}_0^w + \sigma_t \boldsymbol{\epsilon}^w$ and $\mathbf{x}_t^l = \alpha_t \mathbf{x}_0^l + \sigma_t \boldsymbol{\epsilon}^l$ are noisy versions of preferred and dispreferred samples at timestep $t$, $\boldsymbol{\epsilon}_\theta(\cdot, t)$ is the predicted noise, $\beta$ controls the deviation between policy and

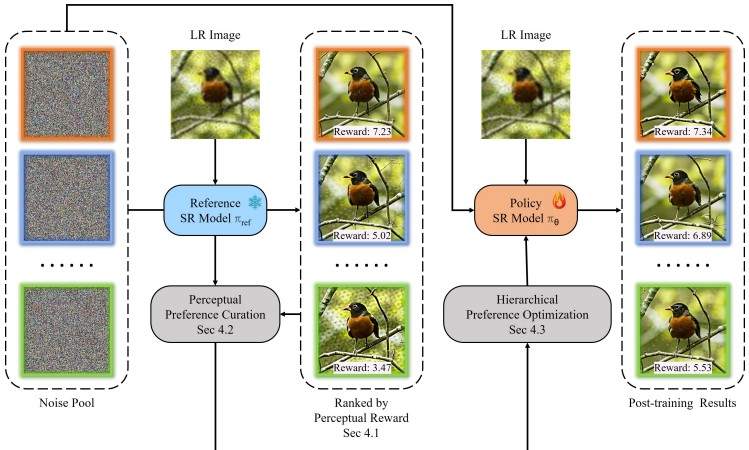

Figure 2: Illustration of our DP²O-SR framework. Given the same LR input and noise, a frozen pre-trained SR model $\pi_{\text{ref}}$ (blue) generates diverse outputs by varying the random seed. These outputs are first evaluated using our proposed perceptual reward (Sec. 4.1), and then filtered through a perceptual preference curation process (Sec. 4.2) to construct pairwise comparisons. A trainable policy model $\pi_\theta$ (orange), initialized from $\pi_{\text{ref}}$, is subsequently optimized via hierarchical preference optimization (Sec. 4.3), which emphasizes informative comparisons and enhances perceptual alignment.

reference models, and $\sigma(\cdot)$ is the sigmoid function normalizing the preference score. For flow-based models, we replace the predicted noise with velocity. This loss encourages better denoising of preferred samples, thereby aligning generation with human preferences.

## 4 Methodology

We propose **DP²O-SR**, a preference-driven optimization framework for Real-ISR using diffusion or flow-based models. Our key insight is to leverage the inherent stochasticity of these models—different noise seeds naturally produce outputs with varying perceptual quality. By scoring these outputs with a human-aligned perceptual reward and forming pairwise preferences, we enable direct supervision of perceptual quality without handcrafted losses or external reward networks. As illustrated in Fig. 2, our framework comprises three components: (1) generating diverse ISR samples from a frozen reference model, (2) ranking them with a perceptual reward to curate preference pairs, and (3) optimizing a trainable policy via uncertainty-aware preference learning. Our design aligns ISR outputs with human perceptual preferences and improves the perceptual quality of both C-SD2 and C-FLUX.

### 4.1 Perceptual Reward Design

Given an LR input $I_{\text{LR}} \in \mathbb{R}^{h \times w \times 3}$, we use a Real-ISR model $\pi_{\text{ref}}$ to generate $M$ ISR candidates $\mathcal{S} = \{I_1, \ldots, I_M\}$ by varying the noise seeds, which often exhibit certain perceptual variations. To rank the perceptual quality of these outputs, we introduce a perceptual reward that aggregates multiple IQA metrics. Specifically, we consider two sets of metrics: a set $\mathcal{FR}$ of FR metrics, which compare outputs against GT images, and a set $\mathcal{NR}$ of NR metrics, which assess quality without reference. Specifically, the $\mathcal{FR}$ set includes LPIPS [58], TOPIQ-FR [7], and AFINE-FR [9], and the $\mathcal{NR}$ set includes MANIQA [51], MUSIQ [19], CLIPIQ+ [41], TOPIQ-NR [7], AFINE-NR [9], and Q-Align [46]. Note that the commonly used distortion-based metrics PSNR and SSIM are excluded, as they are not effective in describing perceptual quality.

For each candidate $I_m, m = 1, ..., M$ and each metric $\phi \in \mathcal{FR} \cup \mathcal{NR}$, we compute a raw score $s_m^\phi$. The scores are first direction-aligned such that higher values indicate better quality. Denote by $s_{\max}^\phi$ and $s_{\min}^\phi$ the maximum and minimum values among the score values of the $M$ candidates, we then normalize each score $s_m^\phi$ as $\bar{s}_m^\phi = (s_m^\phi - s_{\min}^\phi)/(s_{\max}^\phi - s_{\min}^\phi)$. To balance fidelity and perception, we define the perceptual reward of a sample by averaging the normalized scores over the $\mathcal{FR}$ and $\mathcal{NR}$ sets, assigning them equal weight regardless of their sizes:

$$R_m = \frac{0.5}{|\mathcal{FR}|} \sum_{\phi \in \mathcal{FR}} \bar{s}_m^\phi + \frac{0.5}{|\mathcal{NR}|} \sum_{\phi \in \mathcal{NR}} \bar{s}_m^\phi, \tag{2}$$

where $|\cdot|$ denotes the cardinality of a set. Based on this reward, we can identify the top-$N$ and bottom-$N$ candidates to construct preference pairs for training.

## 4.2 Perceptual Preference Data Curation

Given the reward signal, we construct preference pairs by sampling multiple ISR outputs per input using different noise seeds from a single stochastic model. In contrast to Diff-DPO [40], which selects outputs from different models and constructs a single best-versus-worst pair per input, our approach samples $M$ outputs from the same model and ranks them using the reward function $R_m$. We then select the top-$N$ and bottom-$N$ samples as positive and negative sets, respectively, and build $N^2$ possible preference pairs. This allows us to construct richer training signals from a single model without relying on external comparisons or ground-truth.

This design introduces two key control parameters: the number of samples per input ($M$) and the selection ratio ($N/M$). Varying these parameters enables us to control the contrast and diversity of the training signal. For example, smaller values of $N/M$ are expected to yield stronger reward gaps (contrast) between positive and negative samples, potentially accelerating learning. Conversely, larger $N/M$ ratios increase coverage and diversity but may reduce discriminative signal strength.

To explore the effect of these choices across architectures, we apply this curation strategy to two representative models of contrasting capacity: C-SD2 (a 0.8B UNet diffusion model) and C-FLUX (a 12B DiT flow model). We hypothesize that higher-capacity models may benefit from stronger contrast (low $N/M$), whereas smaller models may require more redundancy to ensure stable gradients. The experimental analysis of these hypotheses is provided in Section 5.3.

## 4.3 Hierarchical Preference Optimization

We propose hierarchical preference optimization (HPO), an extension of Diff-DPO that improves preference alignment by adaptively weighting training pairs at two levels: *intra-group* and *inter-group*. While Diff-DPO treats all training pairs equally, HPO focuses on learning from more informative signals by leveraging both local and global variations.

**Intra-group.** Within each group of SR candidates generated from the same LR input, preference pairs $(\boldsymbol{x}_0^w, \boldsymbol{x}_0^l)$ may differ in informativeness. Intuitively, pairs with larger reward gaps $\Delta R = R_w - R_l$ offer stronger supervision. We define the intra-group weight as $w_{\text{intra}}(\boldsymbol{x}_0^w, \boldsymbol{x}_0^l) = |R_w - R_l| + (1 - \mu_{\text{gap}})$, where $\mu_{\text{gap}}$ is the average reward gap over all pairs within the group. This formulation ensures that high-contrast pairs receive greater emphasis while keeping the expected weight normalized.

**Inter-group.** Different LR inputs may yield SR candidate groups with varying levels of perceptual diversity. To prioritize groups that offer stronger supervision, we compute the standard deviation $\sigma_g$ of reward values $\{R_m\}$ within group $g$, and assign inter-group weight as $w_{\text{inter}}(g) = \sigma_g + (1 - \mu_\sigma)$, where $\mu_\sigma$ is the average standard deviation across all groups. This boosts the contribution of more informative groups while keeping the expected group weight close to 1.

**Final Loss Function**. Each training pair $(\boldsymbol{x}_0^w, \boldsymbol{x}_0^l)$ is assigned a total weight $w = w_{\text{intra}}(\boldsymbol{x}_0^w, \boldsymbol{x}_0^l) \cdot w_{\text{inter}}(g)$. The final training objective is $\mathcal{L}_{HPO} = \sum_{(\boldsymbol{x}_0^w, \boldsymbol{x}_0^l)} w \cdot \ell(\boldsymbol{x}_0^w, \boldsymbol{x}_0^l; \theta)$, where $\ell(\cdot)$ denotes the per-pair Diff-DPO loss defined in Eq. 1.

## 5 Experiment

### 5.1 Experimental Settings

**Baselines.** We adopt the ControlNet paradigm [57], where a control branch is trained for the Real-ISR task, using pre-trained Stable Diffusion 2.0 (SD2) [1] and FLUX.1-Dev (FLUX) [20] as backbones. For convenience, we refer to the resulting models as C-SD2 and C-FLUX, respectively, and their improved variants with our method as DP²O-SR (SD2) and DP²O-SR (FLUX). This setup enables us to evaluate DP²O-SR across models that differ substantially in capacity (0.8B vs. 12B), architecture (UNet vs. MMDiT), and generative paradigm (diffusion vs. flow matching). The detailed network architecture of our C-SD2 and C-FLUX can be found in the **Appendix**.

**Evaluation Metrics.** We evaluate our method using a total of 14 IQA metrics, categorized into four groups: (1) *Trained FR metrics*: LPIPS [58], TOPIQ-FR [7], and AFINE-FR [9]; (2) *Trained NR*

Table 1: Performance comparison of different methods. Metric types are categorized into trained FR perceptual (in blue), trained NR perceptual (in green)), untrained NR perceptual (in yellow)), and untrained FR fidelity (in purple)) based metrics. **Red bold** values indicate better performance between the baseline C-SD2, C-FLUX and their boosted versions by DP$^2$O-SR. Arrows indicate whether higher (↑) or lower (↓) values are better.

| Datasets | Metrics | StableSR | DiffBIRv2 | SeeSR | CCSR | AddSR | OSEDiff | C-SD2 | DP$^2$O-SR (SD2) | C-FLUX | DP$^2$O-SR (FLUX) |
|---|---|---|---|---|---|---|---|---|---|---|---|
| *Syn-Test* | LPIPS↓ | 0.4219 | 0.4471 | 0.4322 | 0.4080 | 0.4930 | 0.4043 | 0.4332 | **0.4268** | 0.4260 | **0.4187** |
| | TOPIQ-FR↑ | 0.4208 | 0.4108 | 0.4238 | 0.4357 | 0.3577 | 0.4375 | 0.4336 | **0.4396** | 0.4364 | **0.4489** |
| | AFINE-FR↓ | -0.6309 | -0.9339 | -1.0931 | -0.3396 | -0.7704 | -1.0567 | **-1.1433** | -0.8962 | **-1.0164** | -0.8341 |
| | MANIQA↑ | 0.5707 | 0.6528 | 0.6513 | 0.5956 | 0.7025 | 0.6327 | 0.6684 | **0.7165** | 0.6857 | **0.7199** |
| | MUSIQ↑ | 60.34 | 70.59 | 71.37 | 64.34 | 73.33 | 70.04 | 71.66 | **74.87** | 72.28 | **75.06** |
| | CLIPIQA+↑ | 0.6313 | 0.7491 | 0.7385 | 0.6421 | 0.7742 | 0.7164 | 0.7595 | **0.8124** | 0.7473 | **0.7993** |
| | TOPIQ-NR↑ | 0.5019 | 0.7084 | 0.7098 | 0.5941 | 0.7629 | 0.6341 | 0.7155 | **0.7611** | 0.7019 | **0.7645** |
| | AFINE-NR↓ | -0.8693 | -0.9683 | -1.0483 | -0.7937 | -1.0751 | -0.9879 | -1.0097 | **-1.2263** | -1.2026 | **-1.2764** |
| | QALIGN↑ | 3.2196 | 4.1382 | 4.1614 | 3.3266 | 4.2086 | 3.9801 | 4.2481 | **4.5526** | 4.4266 | **4.7060** |
| | VQ-R1↑ | 3.78 | 4.34 | 4.42 | 3.88 | 4.38 | 4.40 | 4.43 | **4.57** | 4.53 | **4.65** |
| | NIMA↑ | 4.8936 | 5.4096 | 5.3862 | 4.7765 | 5.5352 | 5.2029 | 5.3894 | **5.6417** | 5.4458 | **5.5986** |
| | TOPIQ-IAA↑ | 4.7056 | 5.3929 | 5.3595 | 4.8338 | 5.6047 | 5.1199 | 5.4123 | **5.6106** | 5.3457 | **5.5292** |
| | PSNR↑ | 22.75 | 22.43 | 22.41 | 23.54 | 21.00 | 22.61 | **22.46** | 21.48 | 21.26 | **21.27** |
| | SSIM↑ | 0.5865 | 0.5355 | 0.5648 | 0.5928 | 0.4832 | 0.5775 | **0.5449** | 0.5259 | **0.5158** | 0.5143 |
| *RealSR* | LPIPS↓ | 0.3877 | 0.4288 | 0.3883 | 0.3645 | 0.4539 | 0.3729 | 0.4146 | **0.4045** | **0.4004** | 0.4024 |
| | TOPIQ-FR↑ | 0.4923 | 0.4747 | 0.4881 | 0.5367 | 0.4093 | 0.5059 | **0.4756** | 0.4656 | 0.4824 | **0.4867** |
| | AFINE-FR↓ | -0.7699 | -0.8059 | -0.7439 | -0.9548 | -0.1243 | -0.7174 | **-0.6578** | -0.3331 | -0.5916 | **-0.6097** |
| | MANIQA↑ | 0.6230 | 0.6502 | 0.6451 | 0.6034 | 0.6810 | 0.6335 | 0.6629 | **0.7031** | 0.6632 | **0.6918** |
| | MUSIQ↑ | 65.88 | 69.28 | 69.82 | 63.57 | 71.39 | 69.09 | 70.44 | **73.16** | 69.60 | **72.77** |
| | CLIPIQA+↑ | 0.6501 | 0.7235 | 0.6910 | 0.6216 | 0.7438 | 0.6964 | 0.7295 | **0.7852** | 0.6798 | **0.7571** |
| | TOPIQ-NR↑ | 0.5748 | 0.6760 | 0.6891 | 0.5735 | 0.7262 | 0.6254 | 0.6828 | **0.7429** | 0.6522 | **0.7416** |
| | AFINE-NR↓ | -1.0120 | -0.9860 | -1.0368 | -0.9157 | -1.1449 | -1.0489 | -1.0357 | **-1.1555** | -1.0985 | **-1.0905** |
| | QALIGN↑ | 3.2337 | 3.6866 | 3.6723 | 3.1317 | 3.7625 | 3.6399 | 3.6490 | **4.0206** | 3.6499 | **4.1492** |
| | VQ-R1↑ | 3.78 | 4.12 | 3.98 | 3.80 | 4.08 | 4.09 | 4.08 | **4.21** | 4.02 | **4.32** |
| | NIMA↑ | 4.8150 | 4.9190 | 4.9193 | 4.4545 | 5.1601 | 4.8952 | 4.9914 | **5.2324** | 4.9780 | **5.0740** |
| | TOPIQ-IAA↑ | 4.5856 | 4.8981 | 4.8553 | 4.3509 | 5.0890 | 4.7545 | 4.9238 | **5.1144** | 4.7427 | **4.9798** |
| | PSNR↑ | 24.64 | 24.83 | 25.15 | 26.21 | 23.31 | 25.15 | **23.61** | 22.49 | **23.58** | 23.49 |
| | SSIM↑ | 0.7077 | 0.6500 | 0.7213 | 0.7363 | 0.6404 | 0.7340 | **0.6566** | 0.6500 | **0.6594** | 0.6590 |

*metrics*: MANIQA [51], MUSIQ [19], CLIPIQA+ [41], TOPIQ-NR [7], AFINE-NR [9], and Q-Align [46], which together form the perceptual reward used in training. These two groups constitute the sets $\mathcal{FR}$ and $\mathcal{NR}$ defined in Sec. 4.1 for reward computation. (3) *Untrained NR perceptual metrics*: VQ-R1 [49], NIMA [38], and TOPIQ-IAA [7], used to assess perceptual generalization beyond training targets; (4) *Untrained FR fidelity metrics*: PSNR and SSIM [45], included for completeness. This comprehensive setting allows us to evaluate both in-distribution performance and out-of-distribution generalization of Real-ISR models.

**Finetuning, Post-Training and Testing Datasets.** The C-FLUX model is finetuned for the Real-ISR task using approximately 1 million high-quality images. We use a batch size of 32, a learning rate of $1 \times 10^{-4}$, and train the model for 45,000 steps. The C-SD2 variant follows a similar setup, but is trained with a batch size of 256, a learning rate of $2 \times 10^{-4}$, and 35,000 steps.

To support perceptual post-training, we curate a semantically diverse dataset from the Internet, containing 30,100 high-quality images spanning six major scene types and 266 sub-categories. Among them, 30,000 images are used for post-training, while the remaining 100 images form the *Syn-Test* set. We adopt the first-order degradation pipeline from ResShift [54], which better simulates real-world degradation than the second-order version used in RealESRGAN [43]. We further evaluate generalization performance on real-world benchmarks such as *RealSR* [5].

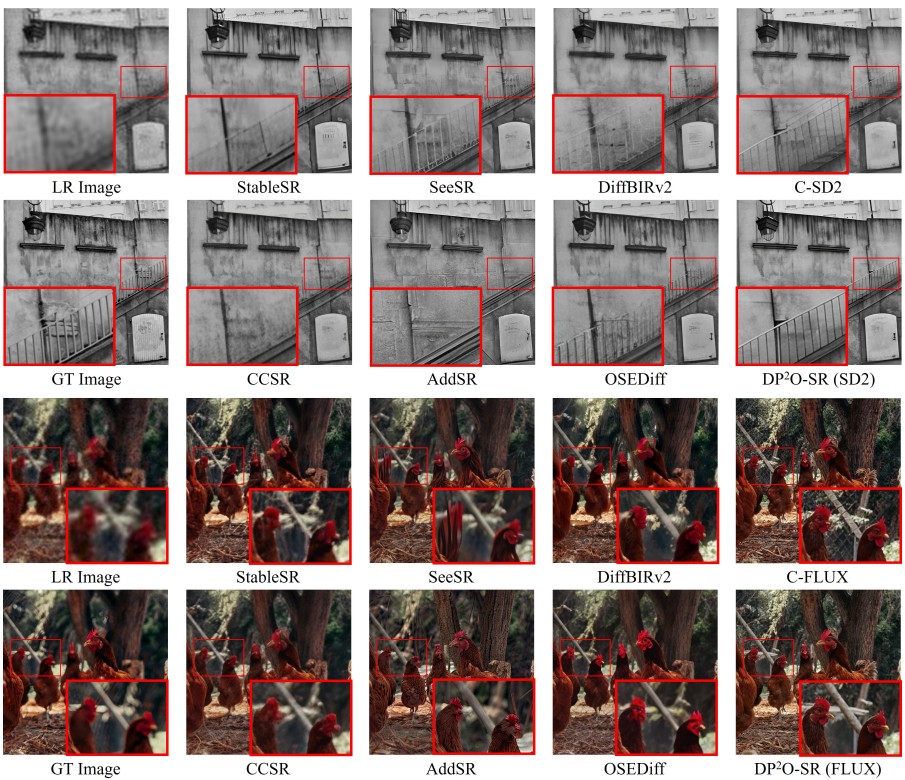

| LR Image | StableSR | SeeSR | DiffBIRv2 | C-SD2 |
| GT Image | CCSR | AddSR | OSEDiff | DP²O-SR (SD2) |
| LR Image | StableSR | SeeSR | DiffBIRv2 | C-FLUX |
| GT Image | CCSR | AddSR | OSEDiff | DP²O-SR (FLUX) |

Figure 3: Qualitative comparisons of different Real-ISR methods. Please zoom in for a better view.

**DP²O-SR Training Configuration.** We train DP²O-SR with a batch size of 1024, a learning rate of $2 \times 10^{-5}$, and set the preference weighting hyperparameter $\beta$ to 5,000. The model is trained for 1,000 iterations. All experiments are conducted on 8×A800 GPUs. To account for the stochasticity of the generative SR models, we sample up to $M = 64$ outputs per LR image during training. Preference pairs are constructed from C-FLUX and C-SD2 outputs, each sampled using 25 and 50 inference steps, respectively. The corresponding classifier-free guidance (CFG) scales are set to 2.5 and 3.5. These settings are kept consistent throughout both training and evaluation.

**Offline Candidate Generation and IQA Labeling.** For offline preference pair construction, we sample 64 outputs per LR image across 30,000 training images using the same 8×A800 GPU setup. This process requires approximately 168 hours for C-SD2 and 432 hours for C-FLUX. The resulting 1.92 million generated images are labeled using our suite of IQA models (Section 4), which takes an additional 72 hours. All configurations are held fixed to ensure reproducibility and fair comparison.

## 5.2 Results on Perceptual Preference Alignment

**Improvement over Baselines.** To evaluate the effectiveness of DP²O-SR, we compare it with the baseline models C-SD2 and C-FLUX using the four categories of metrics. As shown in Tab. 1, our method consistently improves performance in almost all perceptual categories, demonstrating strong alignment with training signals and good generalization beyond them.

On the *Syn-Test* dataset, where the degradation process matches that used during training, we observe clear improvements on trained FR metrics such as LPIPS ($\downarrow 0.4332 \rightarrow 0.4268$) and TOPIQ-FR ($\uparrow 0.4336 \rightarrow 0.4396$) for SD2, along with similar trends on C-FLUX. For trained NR metrics, performance is also enhanced across the board, including MANIQA ($\uparrow 0.6684 \rightarrow 0.7165$), CLIP-IQA+ ($\uparrow 0.7595 \rightarrow 0.8124$), and QALIGN ($\uparrow 4.2481 \rightarrow 4.5526$), indicating that preference-guided fine-tuning effectively aligns outputs with perceptual quality signals. Importantly, our method generalizes well to perceptual metrics not used during training. For example, on the untrained metric VQ-R1, we observe consistent improvements over both baselines (*e.g.*, $4.38 \rightarrow 4.57$ for SD2, $4.40 \rightarrow 4.65$ for FLUX), suggesting that the learned preferences transfer beyond the supervised objectives. This aligns with the well-established perception-distortion tradeoff [3], which suggests that improving perceptual quality often comes at the cost of lower PSNR or SSIM.

**Comparison with SOTA.** Our DP²O-SR models outperform a wide range of state-of-the-art Real-ISR methods on the challenging *RealSR* benchmark, including StableSR [42], DiffBIRv2 [28], SeeSR [48], CCSR [37], AddSR [50], and OSEDiff [47], especially in perceptual metrics. For instance, DP²O-SR (SD2) achieves the highest MANIQA (↑0.7031) and CLIPIQA+ (↑0.7852) among all methods, indicating strong perceptual quality and alignment with human preferences.

In untrained perceptual metrics such as VQ-R1 and NIMA, DP²O-SR (FLUX) also exhibits strong generalization, achieving top-tier results (*e.g.*, VQ-R1 ↑4.32, QALIGN ↑4.1492). These results demonstrate that our method not only boosts perceptual alignment on seen metrics but also generalizes to diverse evaluation criteria beyond the training objectives. This highlights the ability of DP²O-SR to transform moderate diffusion backbones into highly competitive Real-ISR models.

**Qualitative Comparisons.** Fig. 3 presents visual comparisons of different Real-ISR methods. In the first example, the C-SD2 baseline produces dense, irregular stripe artifacts in the region of the staircase. After applying DP²O-SR, these artifacts are effectively removed and replaced with regular clean fence structures, achieving even better reconstruction quality than SeeSR and OSEDiff. In contrast, AddSR and CCSR partially erase the stair details. In the second example, C-FLUX generates grid-like artifacts in the background, which are largely eliminated after applying DP²O-SR. DiffBIRv2 and StableSR fail to reconstruct meaningful facial features, yielding smeared or indistinct results. AddSR distorts the beak geometry, while CCSR over-sharpens edges but lacks semantic accuracy. In addition, some methods tend to hallucinate unnatural chicken heads. In comparison, the results of DP²O-SR (C-FLUX) are visually clean and semantically faithful. It should be noted that both C-SD2 and C-FLUX use the same noise seed before and after applying DP²O-SR. This confirms that our method effectively suppresses semantic artifacts while preserving the overall structure and content of the original generation. Due to space limitations, more visualization results, as well as a user study, can be found in the **Appendix**.

## 5.3 Effect of Sample Count and Selection Ratio

To validate the impact of sample count and selection ratio in our preference curation pipeline, we conduct experiments on two contrasting architectures: C-SD2 and C-FLUX. For each model, we vary the total number of rollouts $M$ and the selection ratio $N/M$. The results reveal several consistent patterns and architecture-specific sensitivities:

**Larger $M$ generally improves performance, but with diminishing returns.** For a fixed $N/M$, increasing $M$ consistently improves training stability and final reward, although the benefit decreases as $M$ increases (*e.g.*, $M = 64$).

**DP²O-SR (C-FLUX) is significantly more stable than DP²O-SR (C-SD2)**. C-SD2 exhibits reward collapse under low $N$ or high $N/M$ settings, likely due to overfitting to sparse or low-contrast supervision. In contrast, C-FLUX maintains stable and monotonic reward curves across most configurations, implying resilience to noisy or weak preference signals.

**Both models exhibit architecture-specific optimal $N/M$ regimes.** For C-SD2, $N/M = 1/4$ yields the best trade-off between stability and reward contrast; lower ratios often lead to collapse, while higher ratios converge more slowly. C-FLUX, on the other hand, performs best at lower ratios such as $1/16$, but suffers degraded performance at $1/2$. This suggests that more capable models can effectively learn from stronger contrast signals, whereas smaller models benefit from greater redundancy and smoother gradients.

These results highlight the importance of tailoring preference curation strategies to model capacity. While moderate $N/M$ ratios consistently perform well, the optimal configuration depends on the model's ability to generalize from noisy or low-contrast supervision. Our approach provides a scalable framework for systematically exploring these trade-offs. Based on these findings, we select $N = 8$ and $M = 32$ for C-SD2, and $N = 4$ and $M = 64$ for C-FLUX in all experiments.

## 5.4 Stochasticity and the Effect of DP²O-SR

Diffusion and flow-based SR models generate diverse outputs due to their inherent stochasticity. To analyze this, we sample $M$ outputs per input and compute perceptual reward statistics: *Best@M*, *Mean@M*, and *Worst@M*. As shown in Fig. 5, for both C-SD2 and C-FLUX baselines, *Mean@M* remains relatively stable as $M$ increases, suggesting that a single sample is generally representative

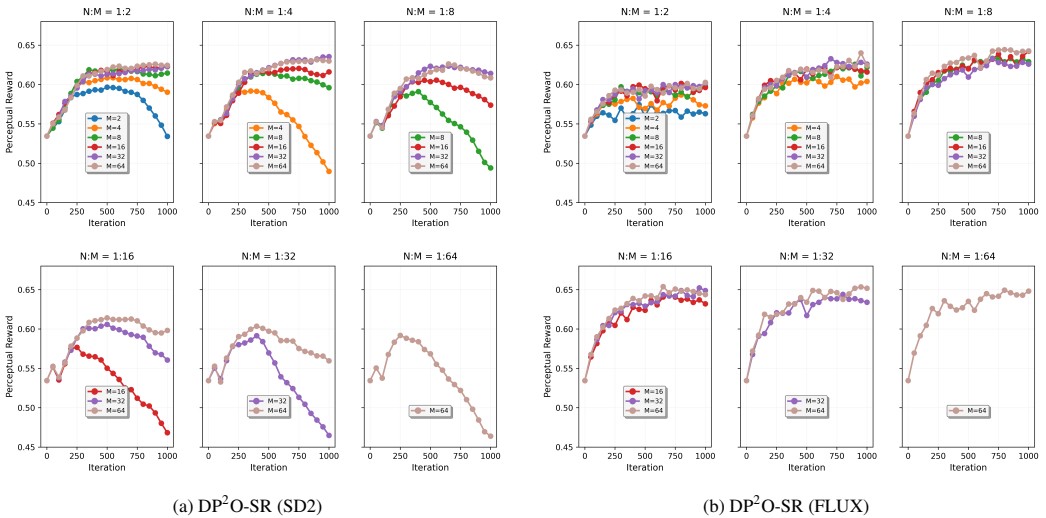

(a) DP$^2$O-SR (SD2)  (b) DP$^2$O-SR (FLUX)

Figure 4: Training curves of DP$^2$O-SR on SD2 and FLUX under varying $M$ and $N/M$ configurations. Larger $M$ generally improves reward, while optimal $N/M$ differs across model capacities.

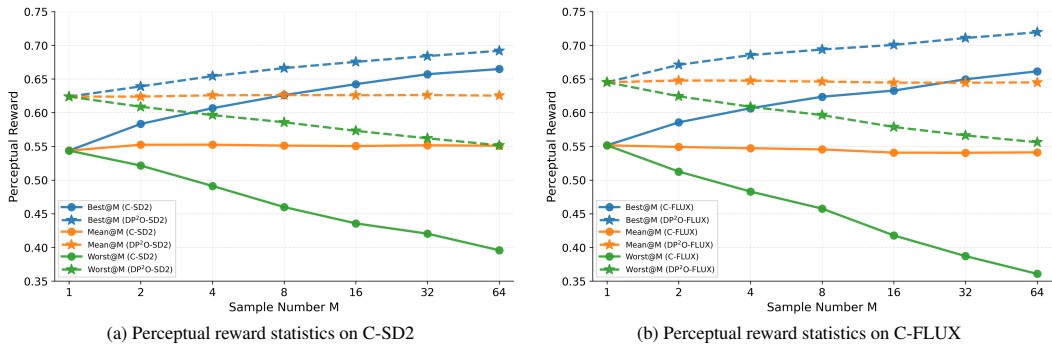

(a) Perceptual reward statistics on C-SD2  (b) Perceptual reward statistics on C-FLUX

Figure 5: Perceptual reward statistics by increasing sample count $M$ on (a) C-SD2 and (b) C-FLUX backbones. In both cases, the baseline models (solid) show a widening quality range with more samples, while DP$^2$O-SR (dashed) consistently improves all statistics—especially *Worst@M*—highlighting stronger robustness and reduced output variability.

of average performance. In contrast, *Best@M* increases and *Worst@M* decreases with larger $M$, indicating increased variability in output quality. DP$^2$O-SR consistently improves all the three statistics across both backbones, with the most notable gain in *Worst@M*. This indicates that our method not only improves average and best-case outcomes, but more importantly, raises the quality floor—leading to more consistent and perceptually robust outputs.

To further assess model stability, we follow the evaluation protocol of CCSR [37], randomly sampling 10 outputs per input and computing the mean and standard deviation of perceptual scores. As shown in Tab. 2, DP$^2$O-SR achieves both better average performance and lower variance across various metrics, confirming its superior robustness and reliability.

## 5.5 Global Reward, Local Refiner

Although our reward function is composed entirely of global IQA metrics, which assess overall image quality, we observe an intriguing behavior: localized refinement. As shown in Fig. 6, we compare the outputs of baseline model C-FLUX and its improved variant DP$^2$O-FLUX across three random seeds, using the same LR input. All inference hyperparameters, including the number of inference steps (25) and classifier-free guidance scale (CFG=2.5), are kept fixed to ensure a controlled comparison. The only varying factor is the random seed. Two noteworthy observations can be made.

Table 2: Performance comparison of different methods on RealSR bench [5].

| Method | LPIPS↓ | TOPIQ-FR↑ | AFINE-FR↓ | MANIQA↑ | MUSIQ↑ | CLIPIQA+↑ | TOPIQ-NR↑ | AFINE-NR↓ | QALIGN↑ |
|---|---|---|---|---|---|---|---|---|---|
| C-SD2 | 0.416±0.018 | 0.473±0.022 | -0.698±0.239 | 0.664±0.019 | 70.34±1.79 | 0.730±0.028 | 0.684±0.034 | -1.032±0.057 | 3.630±0.187 |
| DP$^2$O-SR (SD2) | 0.405±0.009 | 0.467±0.013 | -0.329±0.150 | 0.705±0.012 | 73.24±0.81 | 0.784±0.017 | 0.745±0.015 | -1.157±0.041 | 4.017±0.117 |
| C-FLUX | 0.400±0.023 | 0.480±0.027 | -0.539±0.274 | 0.665±0.025 | 69.70±2.15 | 0.682±0.036 | 0.654±0.046 | -1.096±0.070 | 3.654±0.231 |
| DP$^2$O-SR (FLUX) | 0.403±0.013 | 0.485±0.016 | -0.549±0.176 | 0.694±0.013 | 72.78±0.93 | 0.758±0.019 | 0.743±0.013 | -1.100±0.047 | 4.143±0.113 |

Figure 6: Comparison between C-FLUX and DP$^2$O-FLUX across three random seeds. Trained with global IQA rewards, DP$^2$O-FLUX demonstrates localized refinement (*e.g.*, wing structure), while keeping other regions (*e.g.*, head reflections) almost unchanged.

**Seed Sensitivity in Local Details.** Even within the same model (C-FLUX or DP$^2$O-FLUX), varying the random seed leads to noticeable differences in local structures, such as wing venation and insect head details. This highlights the stochastic nature of diffusion-based generation, especially with under-constrained guidance.

**Local Enhancement from Global Reward.** More surprisingly, under the same seed, DP$^2$O-FLUX often produces visibly sharper and more accurate local structures compared to C-FLUX. For instance, the wing texture (red box) is significantly refined and more faithful to the ground truth, while other regions—such as the specular highlight on the insect's head (green box, white arrow)—remain largely unchanged. This suggests that the model implicitly learns to prioritize perceptually salient regions, even though the reward is computed globally across the entire image.

This phenomenon implies that preference-based training, when guided by global IQA rewards, can lead to localized improvements without any explicit local supervision.

The ablation study on the effectiveness of HPO can be found in the **Appendix**.

## 6 Conclusion

We introduced DP$^2$O-SR, a preference-driven framework for Real-ISR. We defined a perceptual reward that jointly considered naturalness and fidelity. Different from traditional best-vs-worst DPO sampling, we leveraged multiple pairwise comparisons within each group, leading to significantly improved optimization. We further analyzed how the optimal usage of preference samples varied across model scales and proposed a practical hyperparameter optimization strategy to adaptively balance generalization and stability. Extensive experiments demonstrated that our approach brought notable gains over strong baselines, both perceptually and quantitatively.

**Limitations**. DP$^2$O-SR has two main limitations. First, although the IQA-based reward correlates reasonably with human preference, it lacks interpretability and does not fully capture subjective perceptual quality. Designing more accurate and explainable reward models remains an important direction for future work. Second, our current training pipeline is fully offline. Incorporating iterative or online preference optimization may further improve performance and adaptability.

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

# A    Appendix / supplemental material

In this supplementary file, we provide the following materials:

- The detailed network architecture of our C-SD2 and C-FLUX.
- The ablation study on the effectiveness of HPO.
- More visual comparisons.
- Results of user study.

## A.1    The Architecture of C-SD2 and C-FLUX

Fig. 7 illustrates the architectures of the two baseline models, C-SD2 [1] and ControlNet-FLUX [20], both integrated with the proposed DP$^2$O-SR module. The left subfigure shows C-SD2, which is built upon a UNet-based architecture. Its control branch is initialized by duplicating the entire encoder of SD2. Following the standard ControlNet paradigm [57], the low-resolution (LR) condition is first encoded by an image encoder and then added to the input of the frozen pre-trained backbone. The resulting combined features are then processed by the control branch. In C-SD2, the control features are injected into the decoder blocks of the UNet.

The right subfigure presents C-FLUX, which adapts the MMDiT-based [20] architecture. To reduce computational overhead, its control branch is initialized by copying only the first four Double Transformer Blocks from the pre-trained model. Similar to C-SD2, the LR condition is encoded and fused with the model's input before being passed to the control branch. However, in contrast to C-SD2, the control features in C-FLUX are added to the intermediate representations within the Double Transformer blocks, rather than the decoder. This architectural difference reflects the underlying backbone discrepancy between UNet and MMDiT. All pretraining procedures follow the official `diffusers` [39] training recipes to ensure consistency.

## A.2    Effectiveness of HPO

To assess the contributions of the two components in our hierarchical preference optimization (HPO), we conduct an ablation study by separating the effects of intra-group and inter-group weighting. As

Table 3: Ablation study on HPO.

|  | base | intra | inter | both |
| --- | --- | --- | --- | --- |
| Perceptual Reward | 0.645 | 0.648 | 0.649 | **0.651** |

shown in Tab. 3, applying only intra-group weighting—focusing on local contrast within each SR candidate group—already provides a noticeable improvement over the baseline. Similarly, using only inter-group weighting, which emphasizes groups with higher reward dispersion, also leads to better performance. Combining both strategies yields the highest perceptual reward, confirming that the two weighting mechanisms are complementary.

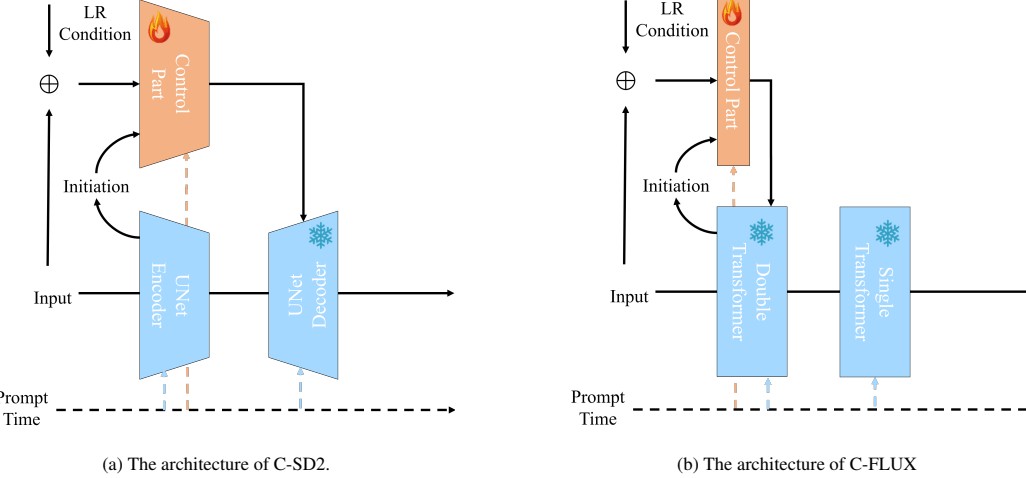

(a) The architecture of C-SD2.          (b) The architecture of C-FLUX

Figure 7: The illustration of baselines (C-SD2 and C-FLUX) equipped with DP$^2$O-SR.

## A.3 More Visual Comparisons

We present additional qualitative comparisons to demonstrate the effectiveness of our proposed method **DP$^2$O-SR** over the two strong baselines: C-SD2 and C-FLUX. For each baseline, we show visual comparisons between it and its improved versions, *i.e.*, C-SD2 vs. DP$^2$O-SR (SD2) and C-FLUX vs. DP$^2$O-SR (FLUX), alongside results from several state-of-the-art methods, including StableSR [42], DiffBIRv2 [28], SeeSR [48], CCSR [37], AddSR [50], and OSEDiff [47].

Below, we first present four representative cases based on the C-SD2 baseline, as shown in Fig. 8. These comparisons highlight the improvements made by DP$^2$O-SR (SD2) in terms of structural fidelity and visual detail, as well as its advantages over existing SOTA methods.

**Case 1: Text Restoration.** In this example, our method demonstrates clear superiority in restoring corrupted text. The baseline C-SD2 fails to recover the correct structure of the letters, especially the letter "m", which is severely distorted. In contrast, DP$^2$O-SR (SD2) produces accurate and sharp letterforms, closely resembling the ground-truth. Notably, the structure of the letter "e" is also cleaner and more distinguishable. While methods like OSEDiff and SeeSR partially recover "e", their outputs remain ambiguous, often appearing between "e" and "o".

**Case 2: Roof Tile Reconstruction.** The second case highlights the reconstruction of architectural details, specifically roof tiles. Although C-SD2 manages to recover a tile-like structure, the spacing and arrangement are overly dense compared to the ground-truth. DP$^2$O-SR (SD2) significantly improves the regularity and spacing of the tiles, resulting in a pattern much more faithful to the original. Other SOTA methods struggle in this case, with most failing to recover the tile layout.

**Case 3: Road Surface Texture.** In the third example, we examine the restoration of road textures and surface patterns. C-SD2 restores the general layout of the path but incorrectly renders the gaps between the bricks as white. Our improved model, DP$^2$O-SR (SD2), not only recovers the brick layout more accurately but also generates clearer and more structurally consistent gap lines. Competing methods mostly fail to reconstruct the pattern or texture meaningfully.

**Case 4: Building Structure Recovery.** The fourth case focuses on complex architectural structures. While C-SD2 recovers a reasonable amount of building details, DP$^2$O-SR (SD2) significantly enriches the reconstruction with finer structural elements and sharper contours. Compared to other methods, the output of DP$^2$O-SR (SD2) shows clear advantages in both structural fidelity and visual realism. Although methods like OSEDiff and DiffBIRv2 generate plausible building forms, their results often include distorted or wavy line structures.

We then present visual comparisons between C-FLUX and DP$^2$O-SR (FLUX), as shown in Fig. 9, to further validate the generality and robustness of our method.

**Case 1: Dice Number Restoration.** In this example, most competing methods fail to reconstruct the dice numbers accurately. While C-FLUX is able to recover the correct "5" on the lower-left white dice, it mistakenly generates "9" on the upper blue and red dice, deviating from the ground truth. In contrast, our DP$^2$O-SR (FLUX) restores the correct "6" patterns with sharp and well-aligned dots, achieving the highest structural fidelity among all methods.

**Case 2: Parrot Eye and Beak Details.** In this case, C-FLUX fails to recover the black eye of the red parrot, resulting in an unnatural facial appearance. DP$^2$O-SR (FLUX) successfully reconstructs the eye region while maintaining a coherent and realistic texture. Although AddSR also restores the eye, it introduces an unnaturally rigid beak that does not exist in the ground-truth. Overall, DP$^2$O-SR (FLUX) provides a more natural and visually consistent reconstruction.

**Case 3: Artifact Removal and Structural Clarity.** For the third example, C-FLUX produces noticeable vertical artifacts that degrade the overall visual quality. Our DP$^2$O-SR (FLUX) effectively suppresses these artifacts while preserving local details and edge sharpness. In comparison, methods such as StableSR, DiffBIRv2, and OSEDiff exhibit a strong "oil-painting" effect, leading to flatter textures and less realistic structures.

**Case 4: Fence Reconstruction.** In the final case, C-FLUX fails to reconstruct the fence structure, resulting in missing geometric elements. DP$^2$O-SR (FLUX) not only restores the complete fence but also aligns well with the ground-truth. Although DiffBIRv2 and OSEDiff generate fence-like patterns, their results appear noisy and irregular. These comparisons further demonstrate that DP$^2$O-SR (FLUX) achieves superior structural accuracy and perceptual realism across diverse scenes.

## A.4 User Study

In the user study, we conduct two categories of experiments. In the first category, we evaluate the effectiveness of $DP^2O$ through pairwise comparisons between the $DP^2O$-enhanced model and its original (non-enhanced) counterpart. In the second category, we perform a multi-way comparison between the $DP^2O$-enhanced model and existing state-of-the-art methods.

In each trial of the experiments, participants are presented with a low-quality (LQ) image along with either two images (for pairwise comparisons) or seven images (for multi-way comparison), and are asked the following question:

> *Given the LQ image, which one of the candidate images is the best high-quality version of it?*

For each backbone model, we invited 10 participants to evaluate 20 distinct scenarios. In each scenario, the participants completed two pairwise comparisons and one multi-way comparison, resulting in 3 evaluations per scenario. This yielded a total of $10 \times 20 \times 2 = 400$ evaluations per backbone. Since the experiments were conducted on two backbones, the entire user study comprised $2 \times 400 = 800$ evaluation outputs.

The results of our user study are summarized in Fig. 10. In the pairwise comparisons (top row), participants consistently preferred the $DP^2O$-enhanced models over the original pre-trained models. Specifically, for C-SD2 (Fig. 10a), 67.5% of selections favor $DP^2O$ over the pre-trained model. For C-FLUX (Fig. 10b), $DP^2O$ outperforms the pre-trained with preference rates of 69.5%.

In the multi-way comparison (bottom row), where participants selected the most plausible high-resolution candidate among seven methods, $DP^2O$ again performs the best. For C-SD2, $DP^2O$ accounts for 28.0% of total votes, ahead of OSEDiff (19.5%) and SeeSR (18.5%). For C-FLUX, $DP^2O$ receives a dominant 44.0% of the votes, significantly outperforming the second-best method, OSEDiff (12%).

These results collectively demonstrate the consistent superiority of $DP^2O$-SR across diverse backbone models and evaluation settings, reflecting its robustness and perceptual quality gains over baselines and state-of-the-art alternatives.

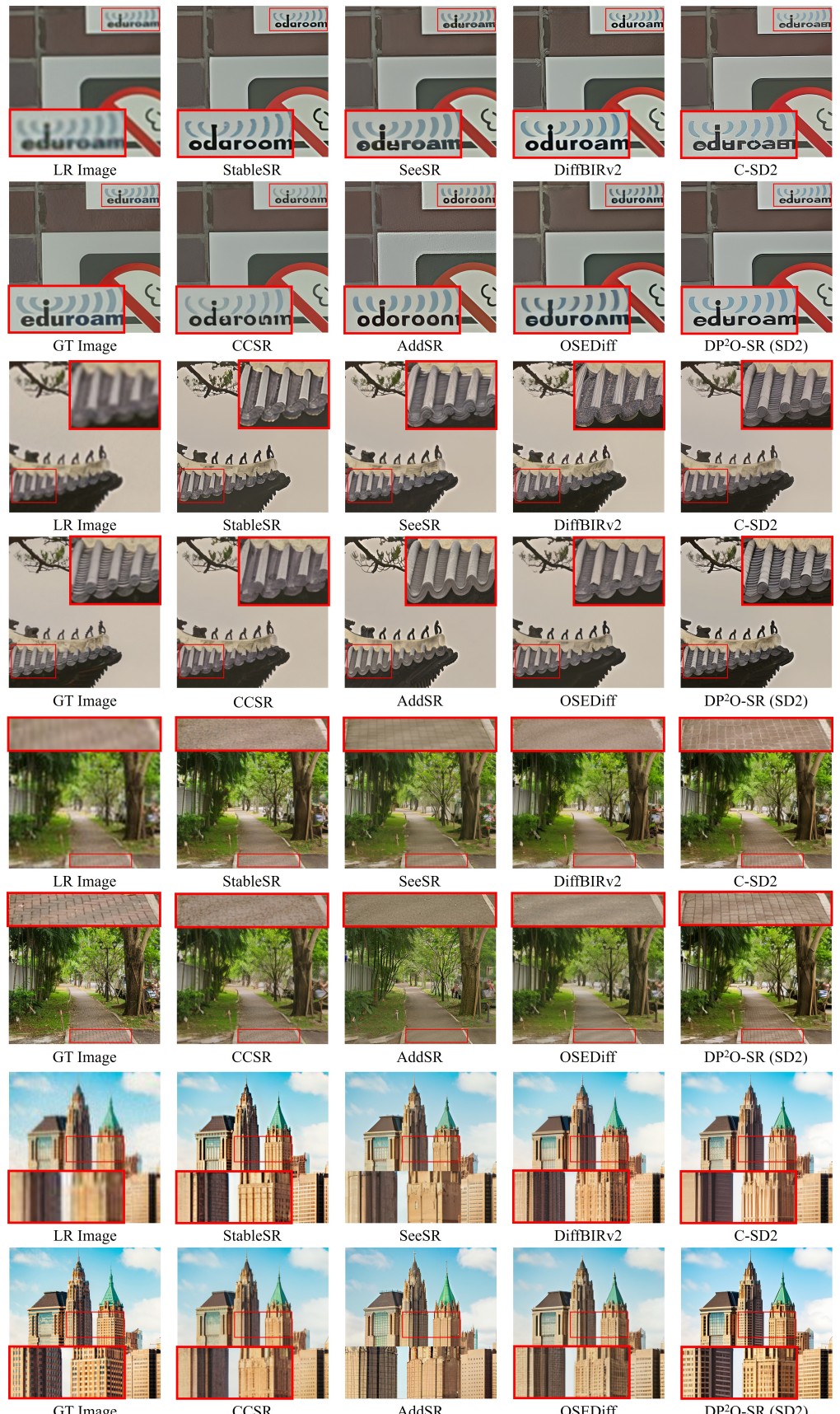

Figure 8: Qualitative comparison between C-SD2, DP²O-SR (SD2), and other Real-ISR methods. Zoom in for visual details.

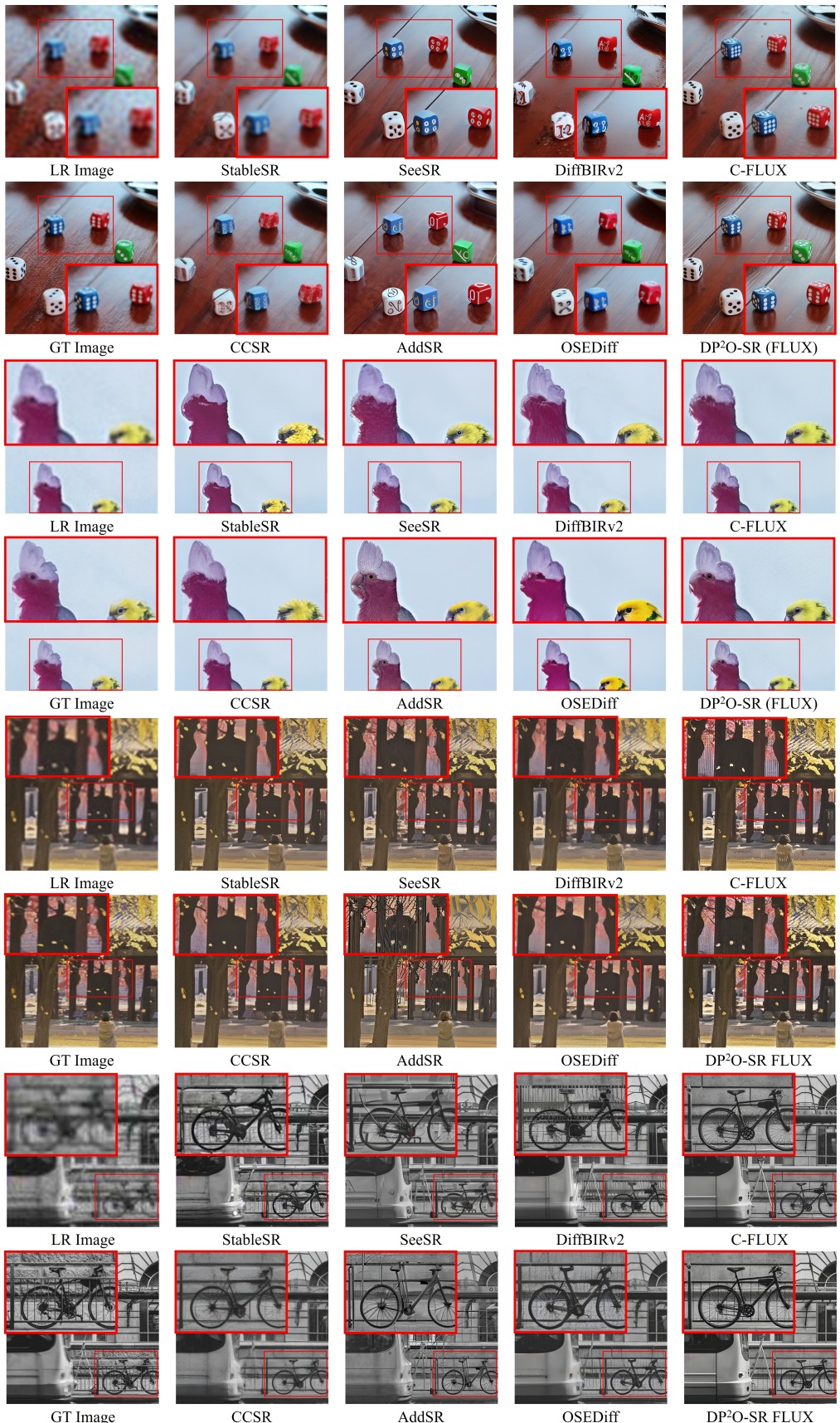

Figure 9: Qualitative comparison between C-FLUX, DP²O-SR (FLUX), and other Real-ISR methods. Zoom in for visual details.

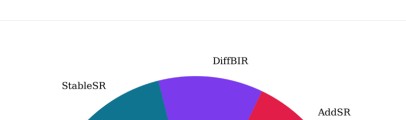

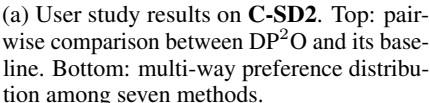

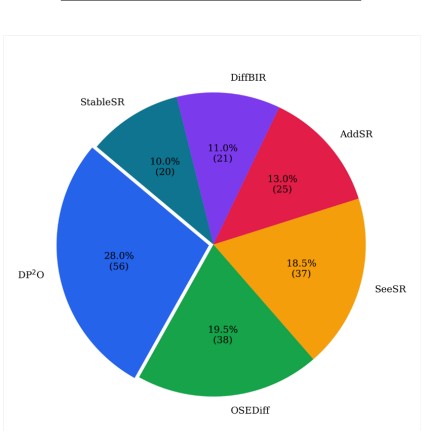

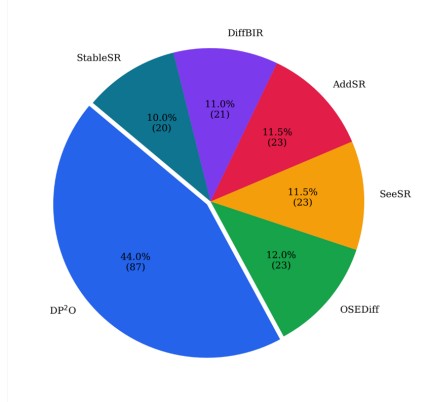

(a) User study results on **C-SD2**. Top: pairwise comparison between DP$^2$O and its baseline. Bottom: multi-way preference distribution among seven methods.

(b) User study results on **C-FLUX**. Top: pairwise comparison between DP$^2$O and its baseline. Bottom: multi-way preference distribution among seven methods.

Figure 10: Summary of user study results on C-SD2 and C-FLUX. In both pairwise and multi-way comparisons, DP$^2$O consistently outperforms baselines and state-of-the-art methods, indicating its strong perceptual advantage in recovering high-quality images from low-resolution inputs.

