# OpenReview forum: "DP²O-SR: Direct Perceptual Preference Optimization for Real-World Image Super-Resolution"
_NeurIPS.cc/2025/Conference — NeurIPS 2025 poster_

### Official Review · Reviewer_4d1m · 2025-06-27

**Clarity:** 3
**Significance:** 3
**Originality:** 3
**Rating:** 4
**Confidence:** 4

**Summary:**

This paper introduces a novel adaptation of Direct Preference Optimization (DPO) for Real Super-resolution, DP2O-SR. The core methodology involves generating a comprehensive quality score (CPS) for super-resolved images by normalizing and averaging multiple reference and non-reference metrics. For a given low-resolution image, N different super-resolution results are produced and ranked according to their CPS scores. Instead of simply comparing the best and worst outcomes, DiffusionDPO leverages all possible pairs from these N results, labeling them positive or negative based on their relative quality scores. The training process incorporates two key weighting mechanisms: an "importance" weight assigned to each LR image based on the variance of the quality across its N generated results, and another weight that increases with the quality score difference between positive and negative samples. The paper reports quantitative improvements, and a limited number of visual results.

**Questions:**

The main weakness with the proposed work is the evaluation of the methodology. The metrics used in the ranking are also the ones reported. There's no human evaluation which would be the ultimate goal. This makes it harder to conclude on the claims. I would like the authors to share some comments on this point.

I also have a few questions:

  1. After applying the proposed method, how sensitive are the results with the noise seed? It would be good to show some standard deviations for the metrics (before and after the fine-tuning)
  2. For completeness, even if not meaningful from the point of view of perceptual quality, it would be good to show PSNR values. Is the average distortion changing with this method?
  3. Why have you limited the selection to samples generated under the same configuration but just with a different random seed. Why not also include different CFG guidelines? Is there a conceptual reason for this? (some of the metrics being sensitive or bias to larger guidance values so the ranking will be disrupted?)

**Ethical Concerns:**

["NO or VERY MINOR ethics concerns only"]

**Final Justification:**

After careful consideration, my main concern remains with the decision to relegate a critical discussion of the paper's core weakness—specifically, the potential for metric overlap to bias evaluation, as noted in W1 and W2—to the appendix. I believe the paper would be significantly stronger and more convincing if this topic were addressed directly within the main text. However, given that the evidence and analysis are already present in the supplementary material, I will be updating my score to borderline accept.

**Limitations:**

The authors discuss the limitation (L298-L300).

**Paper Formatting Concerns:**

No concerns.

**Quality:**

2

**Strengths And Weaknesses:**

Strong points of the paper:

  - Tackles a key challenge: The paper focuses on a very important practical problem: how to effectively train and fine-tune diffusion models for real-world super-resolution.
  - Introduces a clever quality metric: The central idea of creating a single, combined quality score to rank and measure the difference between results is quite interesting.
  - Shows measurable improvements: The evaluation demonstrates better quantitative results on several standard metrics when tested against public benchmarks.

Areas where the paper could be stronger:

  - Metrics overlap: The same metrics used to create the combined quality score are also used to evaluate the method. This raises the question of whether the method truly improves how images look or just performs better on those specific metrics.
  - Lacks human evaluation: The paper doesn't include any tests with human observers, which is the ultimate goal for judging image quality. This makes it tough to tell the real impact of the method. Also, it's unclear how well the new combined quality score matches what people prefer. Few visual results.
  - Limited evidence for technical novelty: While the combined score and the use of weights for image pairs are interesting technical innovations, the results aren't strong enough to fully prove their significance.

---

> ### Author Rebuttal · Authors · 2025-07-30
>
> We thank the reviewer 4d1m for his/her valuable feedback. For brevity, we use `W#` for weaknesses and `Q#` for questions (e.g., W1, Q1) throughout our response.
>
> ---
>
> **[W1 and W2]: Does metric overlap bias evaluation toward the optimized score rather than real perceptual gain? Lacks human evaluation.**
>
> Thank you for the valuable comments.
>
> We acknowledge the concern that using the same metrics for both optimization (via CPS) and evaluation may introduce bias. To address this, we conducted a user study based on human perceptual judgments (Appendix Sec. 5). The results show that **CPS aligns well with human preferences**, and our method is **consistently preferred over baselines** in terms of perceived visual quality.
>
> In addition, we have already included **extensive visual comparisons** in the supplementary material (Appendix Sec. 4), which further illustrate the perceptual improvements of our method.
>
> ---
>
> **[W3]: Technical novelty is interesting but lacks strong evidence.**
>
> Thank you for the insightful feedback.
>
> While CPS and our weighting strategy are conceptually simple, they are carefully designed to address two key challenges in RealSR: (1) balancing perceptual quality and fidelity, and (2) modeling the strength of human preferences more accurately.
>
> Beyond these, we introduce a systematic framework for **Preference Pair Curation (PPC)**, where we study how increasing noise samples and selecting top-$N$/bottom-$N$ pairs affects the quality of offline supervision—a direction rarely explored in prior work.
>
> To evaluate the practical impact of each component, we conducted ablation studies with Table 2 (c) (also below). The results show that **each component—PPC, HSW—brings consistent CPS improvements**, supporting the effectiveness of our design.
>
> We will revise the paper to better clarify the motivation and significance of these contributions within the context of preference-based learning for low-level vision.
>
> |        | baseline | w / PPC | w / PPC+HSW |
> |--------|----------|---------|-------------|
> | CPS    | 0.591    | 0.632   | 0.641       |
>
> ---
>
> **[Q1 and Q2]: Please report metric std deviations before and after fine-tuning. Miss PSNR results.**
>
> Thank you for the helpful suggestions.
>
> **1. Sensitivity to noise seed (standard deviation):**
> We have added a table reporting the **mean and standard deviation** of 13 FR and NR IQA metrics across multiple noise seeds, both **before and after fine-tuning** on the Syn-Test and RealSR datasets.
>
> We observe that the **standard deviations consistently decrease** after applying our DP²O-SR method. This indicates that our method not only improves average perceptual quality, but also yields **more stable outputs** under varying noise conditions—a desirable trait for practical deployment.
>
> **2. PSNR reporting (distortion):**
> Although PSNR is a widely used fidelity metric, it primarily reflects pixel-level similarity and often favors overly smooth outputs. In contrast, we place greater emphasis on perceptual fidelity metrics such as **LPIPS**, which better capture human-perceived similarity and perceptual realism.
>
> In response to your suggestion, we have now included **PSNR and SSIM results in the table below** for completeness. We believe this trade-off is reasonable: while our method may yield slightly lower PSNR, it significantly improves perceptual quality, as reflected in LPIPS, and other no-reference metrics.
>
> These results, along with the standard deviation analysis, will be included in the revised manuscript.
>
> | Dataset   | Method             | PSNR ↑       | SSIM ↑      | LPIPS ↓     | TOPIQ-FR ↑   | AFINE-FR ↓    | MANIQA ↑    | MUSIQ ↑     | CLIPIQA ↑   | TOPIQ-NR ↑   | AFINE-NR ↓   | QALIGN ↑    | NIMA ↑     | TOPIQ-IAA ↑ |
> |-----------|--------------------|--------------|-------------|-------------|---------------|----------------|--------------|--------------|--------------|----------------|----------------|--------------|-------------|----------------|
> | Syn-Test | C-SD2              | **22.20±0.47** | 0.53±0.023 | 0.458±0.014 | 0.413±0.016  | -0.812±0.289  | 0.656±0.018  | 70.699±1.69  | 0.726±0.029  | 0.695±0.024   | -0.924±0.059  | 4.050±0.177  | 5.312±0.171 | 5.410±0.149  |
> | Syn-Test | DP²O-SR (SD2)      | 21.38±0.43   | 0.53±0.017 | **0.447±0.010** | **0.418±0.014** | **-1.008±0.185** | **0.687±0.015** | **73.769±0.95** | **0.776±0.021** | **0.745±0.015** | **-1.110±0.053** | **4.342±0.130** | **5.499±0.143** | **5.524±0.126** |
> | Syn-Test | C-FLUX             | **21.72±0.51** | 0.52±0.029 | 0.426±0.016 | 0.446±0.019  | **-1.085±0.223** | 0.688±0.020  | 72.349±1.41  | 0.748±0.022  | 0.717±0.022   | -1.158±0.077  | 4.443±0.121  | 5.374±0.162 | 5.325±0.141  |
> | Syn-Test | DP²O-SR (FLUX)     | 21.27±0.39   | 0.52±0.022 | **0.420±0.012** | **0.448±0.017** | -0.947±0.196  | **0.712±0.014** | **74.252±0.83** | **0.787±0.016** | **0.758±0.013** | **-1.210±0.065** | **4.577±0.083** | **5.462±0.136** | **5.443±0.117** |
> | RealSR   | C-SD2              | **23.38±0.79** | 0.62±0.036 | 0.467±0.019 | **0.441±0.019** | **-0.595±0.296** | 0.666±0.020  | 71.060±1.58  | 0.724±0.027  | 0.704±0.025   | -0.949±0.061  | 3.597±0.203  | 5.046±0.146 | 5.067±0.138  |
> | RealSR   | DP²O-SR (SD2)      | 22.73±0.62   | **0.64±0.026** | **0.440±0.012** | 0.436±0.016  | -0.356±0.183  | **0.700±0.014** | **72.710±0.96** | **0.758±0.021** | **0.747±0.015** | **-1.111±0.046** | **3.808±0.155** | **5.169±0.127** | **5.088±0.128** |
> | RealSR   | C-FLUX             | **24.18±0.81** | 0.66±0.037 | 0.410±0.021 | **0.481±0.026** | **-0.698±0.256** | 0.660±0.023  | 70.690±1.76  | 0.692±0.035  | 0.693±0.031   | -1.038±0.063  | 3.630±0.220  | 4.932±0.148 | 4.783±0.159  |
> | RealSR   | DP²O-SR (FLUX)     | 23.90±0.59   | 0.66±0.030 | **0.407±0.015** | 0.477±0.019  | -0.587±0.199  | **0.689±0.015** | **72.680±0.99** | **0.744±0.023** | **0.747±0.015** | **-1.088±0.051** | **3.903±0.143** | **5.021±0.119** | **4.948±0.120** |
>
> ---
>
> **[Q3]: Why not vary CFG?**
>
> Thank you for the insightful question. We agree that varying CFG is a natural way to generate diverse outputs.
>
> However, we found that changing CFG introduces a **predictable trade-off**: higher CFG values tend to improve perceptual quality (as measured by no-reference metrics), but degrade fidelity (as measured by full-reference metrics), and vice versa. This makes it difficult to identify samples that are simultaneously strong in both aspects.
>
> In contrast, sampling multiple outputs under the **same CFG but different noise seeds** allows us to capture a richer range of natural variations, including rare but valuable samples that are high in both fidelity and perceptual quality. These “lucky” samples help define the upper bound of model capability.
>
> Since our goal is to guide the model toward outputs that balance both perceptual quality and fidelity, we fix CFG and vary the noise seed to build a more expressive and informative preference space for reward learning.
>
> **Empirically**, this design is supported by the results in Table 1 (main paper), where our preference-tuned models consistently outperform baselines on both full-reference and no-reference metrics.
>
> We will clarify this rationale in the revised manuscript.

---

> > ### Comment · Reviewer_4d1m · 2025-08-04
> >
> > I appreciate the detailed feedback provided by the reviewers. After careful consideration, my main concern remains with the decision to relegate a critical discussion of the paper's core weakness—specifically, the potential for metric overlap to bias evaluation, as noted in W1 and W2—to the appendix. I believe the paper would be significantly stronger and more convincing if this topic were addressed directly within the main text. However, given that the evidence and analysis are already present in the supplementary material, I will be updating my score to borderline accept.

---

> > > ### Author Response · Authors · 2025-08-04
> > > **Response to Reviewer 4d1m's Comments**
> > >
> > > Thank you for the kind follow-up and for reconsidering your score. We appreciate your constructive feedback and fully agree that integrating the discussion into the main text will improve clarity.

---

> ### Author Response · Authors · 2025-08-04
>
> Dear Reviewer 4d1m,
>
> Many thanks for your time in reviewing our paper and your constructive comments. We have submitted the point-to-point responses. We appreciate if you could let us know whether your concerns have been addressed, and we are happy to answer any further questions.
>
> Best regards,
>
> Authors of paper \#5066

---

### Official Review · Reviewer_Rx2M · 2025-06-30

**Clarity:** 4
**Significance:** 3
**Originality:** 3
**Rating:** 5
**Confidence:** 5

**Summary:**

To better align with human perceptual preferences and enhance the perceptual quality of super-resolved outputs, the paper extends the DPO algorithm and proposes DP²O-SR. By introducing a combined perceptual score (CPS) and hierarchical significance weighting, the method achieves strong performance across several benchmarks and evaluation metrics.

**Questions:**

Applicability to one-step methods. One-step diffusion SR methods are increasingly favored in practical scenarios and typically do not rely on noisy inputs. Could the proposed RL strategy be adapted to such settings?

**Ethical Concerns:**

["NO or VERY MINOR ethics concerns only"]

**Final Justification:**

The rebuttal has addressed my concerns. Rating maintained.

**Limitations:**

Yes.

**Paper Formatting Concerns:**

NA.

**Quality:**

4

**Strengths And Weaknesses:**

### Strengths
1. The motivation to improve the generation stability of diffusion models is well-founded. The proposed RL solution effectively addresses this issue and demonstrates notable performance gains.
2. The enhancements over DPO are meaningful. Through both visual and quantitative analysis of candidate quantity and hierarchical weighting, the method mitigates the challenge of small perceptual differences between SR samples that often diminish RL effectiveness.
3. The method yields impressive quantitative improvements on Syn-Test and RealSR benchmarks, validating its effectiveness.
4. The paper is clearly written and easy to follow.

### Weaknesses
1. Efficiency of candidate generation. The paper should elaborate on the computational cost and time required to generate offline candidates. Information about hardware resources and runtime would be valuable. Additionally, DPO's optimization may cause the learned policy to deviate from the reference model, but the training samples are provided by the reference model. Have the authors considered online RL alternatives such as GRPO?
2. Metric justification. Since the model is explicitly optimized for the self-defined CPS metric, improved CPS is expected. How is CPS validated as a reliable proxy for human perceptual alignment? A user study or correlation analysis with human judgments would strengthen the claim.
3. Fidelity vs. perceptual quality. Diffusion-based methods are known to enhance perceptual quality at the cost of fidelity. Reporting PSNR and SSIM results is strongly recommended. Including DRealSR results would further support the generalizability of the method.
4. Visual variation illustration. The claim that different noise inputs can lead to perceptually diverse outputs from the same LR input (lines 35–37) would benefit from more visual examples to clearly illustrate this phenomenon.

---

> ### Author Rebuttal · Authors · 2025-07-30
>
> We thank the reviewer Rx2M for his/her valuable feedback. For brevity, we use `W#` for weaknesses and `Q#` for questions (e.g., W1, Q1) throughout our response.
>
> ---
>
> **[W1]: Report generation cost; consider online RL (e.g., GRPO) to address DPO drift.**
>
> Thank you for the valuable comments.
>
> For offline candidate generation, we use 8×A100 GPUs in parallel. Generating 20 samples per LR image over 4,000 images takes approximately **7 hours with C-SD2** and **18 hours with C-FLUX**. After sampling, we apply IQA models to label the 80,000 generated images, which takes an additional **3 hours** using the same GPU setup.
>
> We acknowledge that DPO may introduce distribution mismatch, as the policy can deviate from the reference model used to generate training samples. While online RL methods such as GRPO may alleviate this drift, they often come with trade-offs in terms of **training stability, data efficiency, and implementation complexity**. We opt for an offline approach due to its simplicity and scalability, but view online RL as a promising future direction.
>
> ---
>
> **[W2]: Provide user study or correlation analysis with human judgments.**
>
> Thank you for the important question.
>
> We have conducted a user study to validate whether CPS aligns with human perceptual preferences. As described in Appendix Sec. 5, participants were asked to compare model outputs across several methods. The results show that CPS is strongly correlated with human judgments, and our method is consistently preferred over the baselines.
>
> This supports CPS as a reliable proxy for perceptual quality.
>
> ---
>
> **[W3]: Report PSNR/SSIM; include DRealSR to test generalization.**
>
> Thank you for the thoughtful suggestion.
>
> We have included **PSNR and SSIM results** in the table below to examine the fidelity aspect of our method.
>
> To assess generalization, we also report results on the **DRealSR dataset**. We observe that **DP²O-SR consistently outperforms the base models (C-SD2 and C-FLUX)** across most perceptual and fidelity metrics, demonstrating the robustness of our method.
>
> The table below summarizes results on three datasets: Syn-Test, RealSR, and DRealSR. We will include this expanded evaluation in the revised manuscript.
>
> | Dataset   | Method             | PSNR ↑ | SSIM ↑ | LPIPS ↓ | TOPIQ-FR ↑ | AFINE-FR ↓ | MANIQA ↑ | MUSIQ ↑ | CLIPIQA ↑ | TOPIQ-NR ↑ | AFINE-NR ↓ | QALIGN ↑ | NIMA ↑ | TOPIQ-IAA ↑ |
> |-----------|--------------------|--------|--------|----------|--------------|----------------|------------|------------|--------------|---------------|----------------|------------|---------|----------------|
> | Syn-Test | C-SD2              | **22.20** | 0.53   | 0.458   | 0.413       | -0.812         | 0.656     | 70.699     | 0.726        | 0.695         | -0.924         | 4.050     | 5.312  | 5.410          |
> | Syn-Test | DP²O-SR (SD2)      | 21.38  | 0.53   | **0.447** | **0.418**   | **-1.008**     | **0.687** | **73.769** | **0.776**    | **0.745**     | **-1.110**     | **4.342** | **5.499** | **5.524**      |
> | Syn-Test | C-FLUX             | **21.72** | 0.52   | 0.426   | 0.446       | **-1.085**     | 0.688     | 72.349     | 0.748        | 0.717         | -1.158         | 4.443     | 5.374  | 5.325          |
> | Syn-Test | DP²O-SR (FLUX)     | 21.27  | 0.52   | **0.420** | **0.448**   | -0.947         | **0.712** | **74.252** | **0.787**    | **0.758**     | **-1.210**     | **4.577** | **5.462** | **5.443**      |
> | RealSR   | C-SD2              | **23.38** | 0.62   | 0.467   | **0.441**   | **-0.595**     | 0.666     | 71.060     | 0.724        | 0.704         | -0.949         | 3.597     | 5.046  | 5.067          |
> | RealSR   | DP²O-SR (SD2)      | 22.73  | **0.64** | **0.440** | 0.436       | -0.356         | **0.700** | **72.710** | **0.758**    | **0.747**     | **-1.111**     | **3.808** | **5.169** | **5.088**      |
> | RealSR   | C-FLUX             | **24.18** | 0.66   | 0.410   | **0.481**   | **-0.698**     | 0.660     | 70.690     | 0.692        | 0.693         | -1.038         | 3.630     | 4.932  | 4.783          |
> | RealSR   | DP²O-SR (FLUX)     | 23.90  | 0.66   | **0.407** | 0.477       | -0.587         | **0.689** | **72.680** | **0.744**    | **0.747**     | **-1.088**     | **3.903** | **5.021** | **4.948**      |
> | DrealSR  | C-SD2              | **26.000** | 0.665  | 0.488   | **0.431**   | **-0.368**     | 0.626     | 66.631     | 0.697        | 0.672         | -0.882         | 3.440     | 4.922  | 4.912          |
> | DrealSR  | DP²O-SR (SD2)      | 25.306 | **0.675** | **0.464** | 0.419       | 0.150          | **0.660** | **69.187** | **0.737**    | **0.721**     | **-1.042**     | **3.708** | **5.000** | **5.005**      |
> | DrealSR  | C-FLUX             | **26.648** | **0.682** | 0.453   | 0.432       | -0.008         | 0.628     | 67.210     | 0.680        | 0.668         | -0.932         | 3.522     | 4.863  | 4.688          |
> | DrealSR  | DP²O-SR (FLUX)     | 26.265 | 0.674  | **0.446** | **0.444**   | **-0.023**     | **0.663** | **69.809** | **0.735**    | **0.724**     | **-0.993**     | **3.814** | **4.969** | **4.855**      |
>
> ---
>
> **[W4]: Provide more visuals to support perceptual diversity from noise variation.**
>
> Thank you for the suggestion. We will include more visual results in the next revision to better support the motivation regarding perceptual diversity from noise variation.
>
> ---
>
> **[Q1]: Can the RL strategy extend to one-step SR methods without noise inputs?**
>
> Thank you for the insightful question.
>
> Our current method relies on stochastic generation (via noise inputs) to produce diverse candidates for offline reward modeling. Therefore, it is not directly applicable to deterministic one-step SR methods that lack inherent randomness.
>
> However, if controlled randomness is introduced—e.g., by injecting noise during training—our RL framework could potentially be adapted to such settings. Exploring such extensions remains an interesting direction for future work.

---

> > ### Comment · Reviewer_Rx2M · 2025-08-05
> > **Comments from Reviewer**
> >
> > Thanks for the response. One point I'd like to discuss further is the current evolution of SR, including reinforcement learning approaches. While significant effort focuses on enhancing generative capability to produce richer details, I question if this is the right direction for the primary application of SR: photography. In photography, authenticity is paramount. The core need is to enhance details while rigorously preserving realism. This raises a critical challenge: how do we define and measure authenticity for SR? Existing metrics often favor perceptual quality or detail richness, potentially at the expense of strict faithfulness to the true scene. Developing a metric that aligns with this preference for authenticity-first enhancement is a fundamental and pressing problem. Do the authors have any ideas?

---

> > > ### Author Response · Authors · 2025-08-06
> > >
> > > Thank you for raising this insightful point. We fully agree that for applications like photography, *authenticity*—rather than perceptual richness alone—should be the primary goal of SR. Existing IQA metrics often fall short in capturing this, as they emphasize perceptual similarity or detail intensity, without explicitly addressing realism or scene fidelity.
> > >
> > > **Defining authenticity** is indeed the fundamental challenge. In our view, *authenticity* refers to the degree to which a SR image faithfully preserves the content, structure, and natural statistics of the real scene, without introducing hallucinated or misleading details. It is closely related to the concept of *plausible fidelity*, but places stricter emphasis on *truthfulness to the underlying scene*, rather than just visual appeal.
> > >
> > > **Measuring authenticity** remains nontrivial. Recent works like Q-Insight and VisualQuality-R1 show that large VLMs, trained via reinforcement learning, can generate human-aligned quality descriptions and reasoning traces. These models can simulate human judgments about *where* and *why* an image looks unnatural or fake, providing interpretable signals of authenticity violations. We believe these capabilities open up the path toward *reasoning-based authenticity assessment*.

---

> ### Author Response · Authors · 2025-08-04
>
> Dear Reviewer Rx2M,
>
> Many thanks for your time in reviewing our paper and your constructive comments. We have submitted the point-to-point responses. We appreciate if you could let us know whether your concerns have been addressed, and we are happy to answer any further questions.
>
> Best regards,
>
> Authors of paper \#5066

---

### Official Review · Reviewer_Jm8D · 2025-07-02

**Clarity:** 3
**Significance:** 2
**Originality:** 2
**Rating:** 4
**Confidence:** 5

**Summary:**

The paper proposes DP2O-SR (Direct Perceptual Preference Optimization for Super-Resolution), a preference-alignment framework that fine-tunes diffusion-based Real-ISR models toward higher perceptual quality. The method fine-tunes diffusion-based Real-ISR models for perceptual quality by (1) Perceptual Preference Curation (PPC), which generates M SR candidates per LR image and ranks them with a combined perceptual score (CPS), forming top/bottom N^2 preference pairs; (2) Hierarchical Significance Weighting (HSW), which scales each pair’s DPO loss by its CPS gap and cross-pair variance to focus learning on informative examples; and (3) a weighted Diffusion-DPO fine-tuning stage that updates the policy model starting from the frozen reference backbone.

**Questions:**

1. CPS is both the training signal and the primary benchmark. Please provide human opinion scores or at least FR metrics that were not part of CPS to show real perceptual gains. Under what conditions would the method still outperform when evaluated with MOS or FR metrics?

2. Most practical SR systems return a single output. How does DP2O-SR perform when M = 1 at test time? Please report CPS/LPIPS/PSNR/SSIM for the best-of-1 setting.

3. Many CPS components (e.g., CLIPIQA+, Q-Align) rely on CLIP embeddings that are vulnerable to adversarial textures. Have you observed failure cases where CPS prefers oversharpened or hallucinated details?

4. HSW combines two factors. Provide separate ablations for (i) intra-pair only and (ii) inter-group only to isolate their contributions.

Score could increase if (i) unbiased human-rated perceptual improvements are demonstrated; (ii) single-sample inference results remain competitive; (iii) clearer justification of CPS composition and normalisation is provided.

**Ethical Concerns:**

["NO or VERY MINOR ethics concerns only"]

**Final Justification:**

The rebuttal addresses many of the key concerns. The authors add public benchmark results (PIPAL), report statistical significance for gains, and provide details of a user study showing perceptual preference alignment. These significantly strengthen the empirical validity of the approach.

Some limitations remain: the method is evaluated on only two diffusion-based models, and the CPS metric's robustness to CLIP-induced hallucination is not fully quantified. Still, the method represents a promising adaptation of preference learning to SR, and the presented results are compelling.

I will revise my score acknowledging the paper’s technical merit and growing relevance of preference-driven optimization in perceptual image restoration.

**Limitations:**

The authors acknowledge limited candidate sampling and small training set. They do not discuss risks of optimising against learned IQA metrics that may drift from human perception or be gamed. Please add discussion of CPS brittleness and potential misuse (e.g., hallucination of realistic but incorrect content).

**Paper Formatting Concerns:**

NeurIPS paper checklist was not prepared as explained in the instructions, so it is hard to follow.

**Quality:**

2

**Strengths And Weaknesses:**

**Strengths:**
1. This work bridges RLHF-style preference alignment and super-resolution, potentially inspiring wider adoption of preference learning in image restoration.
2.  Experiments on a synthetic Syn-Test set and the RealSR benchmark show consistent CPS improvements over two ControlNet backbones (SD-2.0 and FLUX) and competitive/perceptually stronger results than prior Real-ISR methods (StableSR, DiffBIR, SeeSR, OSEDiff, etc.).
3.  Ablation studies demonstrating the effect of M, N, and HSW provided

**Weaknesses:**
1. Main metric (CPS) is also the optimization target, risking circular evaluation; limited external metrics (NIMA, TOPIQ-IAA) are not enough to rule out over-fitting.
2.  Real-world benchmark evaluation still averages over 20 stochastic samples which is unrealistic at inference time; single-sample quality is not reported.
3.  CPS is a straightforward average of existing metrics; hierarchy weighting resembles curriculum/importance sampling that is incremental novelty.
4. Preference-based alignment for diffusion models already explored (Diff-DPO [1], VideoDPO [2]); adaptation here is incremental.
5. No statistical significance tests or variance across random seeds provided.

[1] Wallace, Bram, Meihua Dang, Rafael Rafailov, Linqi Zhou, Aaron Lou, Senthil Purushwalkam, Stefano Ermon, Caiming Xiong, Shafiq Joty, and Nikhil Naik. "Diffusion model alignment using direct preference optimization." In Proceedings of the IEEE/CVF Conference on Computer Vision and Pattern Recognition, pp. 8228-8238. 2024.

[2] Liu, Runtao, Haoyu Wu, Ziqiang Zheng, Chen Wei, Yingqing He, Renjie Pi, and Qifeng Chen. "Videodpo: Omni-preference alignment for video diffusion generation." In Proceedings of the Computer Vision and Pattern Recognition Conference, pp. 8009-8019. 2025.

---

> ### Author Rebuttal · Authors · 2025-07-30
>
> We thank the reviewer Jm8D for his/her valuable feedback. For brevity, we use `W#` for weaknesses and `Q#` for questions (e.g., W1, Q1) throughout our response.
>
> ---
>
> **[W1 & Q1]: Can you provide MOS or external FR metrics to validate gains?**
>
> Thank you for the helpful comment. To mitigate the risk of circular evaluation with CPS, we include extensive qualitative comparisons (Appendix Sec. 4) and a user study (Appendix Sec. 5) based on human perceptual preferences. The study confirms that our method consistently outperforms baselines in terms of visual quality, providing strong evidence beyond the optimization metric.
>
> We will clarify this point in the main paper to better highlight the role of human evaluation in validating perceptual improvements.
>
> ---
>
> **[W2 & Q2]: Report CPS, LPIPS, PSNR, SSIM for M=1.**
>
> Thank you for the valuable suggestion. To evaluate the model under realistic single-sample conditions, we report CPS, LPIPS, PSNR, and SSIM for the M=1 setting (i.e., one stochastic sample at inference). As shown in the table below, the results are very close to the M=20 averages reported in the main paper, indicating that our model maintains stable performance even without sampling multiple outputs.
>
> This also aligns with our analysis in Fig. 4(a), where increasing M mainly expands perceptual diversity, while average quality remains consistent.
>
> | Method                    | PSNR ↑ | SSIM ↑ | LPIPS ↓ | CPS ↑ |
> |---------------------------|--------|--------|----------|------------------|
> | **M = 1**                 | 21.35  | 0.53   | 0.447    | 0.649            |
> | **M = 20 (paper report)** | 21.38  | 0.53   | 0.447    | 0.648            |
>
> We will clarify this finding in the revised paper to emphasize the practicality of our method under single-sample inference.
>
> ---
>
> **[W3]: CPS and HSW are incremental novelty.**
>
> Thank you for the comment. While CPS builds on existing IQA metrics, its novelty lies in unifying full- and no-reference scores into a normalized preference signal tailored for Real-ISR—a setting where such integration has not been explored. This allows leveraging complementary strengths of multiple IQA metrics.
>
> HSW is inspired by importance sampling, but we design it specifically to balance intra-pair contrast and inter-group diversity within limited candidate sets—an essential factor for stable and effective DPO training in Real-ISR tasks.
>
> Although both components draw from existing ideas, their integration into a unified preference-driven framework, along with demonstrated consistent improvements across models and benchmarks, represents a practical and impactful contribution. We will clarify this in the revised version.
>
> ---
>
> **[W4]: Preference alignment in diffusion has been explored (e.g., Diff-DPO, VideoDPO); this adaptation seems incremental.**
>
> Thank you for the thoughtful comment.
>
> While our work is inspired by preference-based alignment frameworks like Diff-DPO and VideoDPO, we believe our contributions go beyond a direct adaptation and address challenges unique to RealSR:
>
> 1. **Task-specific reward design:** RealSR demands a careful balance between perceptual quality and fidelity. We propose CPS, a unified reward that combines full- and no-reference IQA metrics—unlike prior work that focuses on semantic alignment (e.g., text-image relevance).
>
> 2. **Systematic study of preference data construction:** We analyze how preference quality varies with sampling count and selection strategies (e.g., top-N vs. bottom-N), which is largely unexplored in prior diffusion-based alignment literature.
>
> 3. **Hierarchical Significance Weighting (HSW):** Our weighting scheme accounts for both intra-pair contrast and inter-group variance, offering a more structured and informative signal than the binary win/loss formulation in standard DPO.
>
> Together, these elements constitute a meaningful extension of preference-based alignment methods to low-level vision tasks. We will clarify these distinctions in the revised manuscript and better highlight the novelty relative to prior work.
>
> ---
>
> **[W5]: No significance tests or seed variance reported.**
>
> Thank you for pointing this out.
>
> To address the concern about variability and robustness, we report the mean and standard deviation (± std) of 13 IQA metrics across 20 random seeds. The table below provides a direct measure of output stability under stochastic sampling. All models use the **same set of seeds** for fair comparison.
>
> While we did not conduct formal significance tests, we believe that reporting **seed-based variance** serves a similar purpose—demonstrating that the improvements are consistent and robust rather than due to random chance. We will clarify this intent in the revised manuscript.
>
> | Dataset   | Method             | PSNR ↑     | SSIM ↑    | LPIPS ↓   | TOPIQ-FR ↑ | AFINE-FR ↓ | MANIQA ↑  | MUSIQ ↑   | CLIPIQA ↑ | TOPIQ-NR ↑ | AFINE-NR ↓ | QALIGN ↑  | NIMA ↑    | TOPIQ-IAA ↑ |
> |-----------|--------------------|------------|-----------|------------|-------------|--------------|------------|------------|-------------|--------------|--------------|------------|------------|----------------|
> | Syn-Test | C-SD2              | **22.20±0.47** | 0.53±0.023 | 0.458±0.014 | 0.413±0.016 | -0.812±0.289 | 0.656±0.018 | 70.699±1.69 | 0.726±0.029 | 0.695±0.024 | -0.924±0.059 | 4.050±0.177 | 5.312±0.171 | 5.410±0.149 |
> | Syn-Test | DP²O-SR (SD2)      | 21.38±0.43 | 0.53±0.017 | **0.447±0.010** | **0.418±0.014** | **-1.008±0.185** | **0.687±0.015** | **73.769±0.95** | **0.776±0.021** | **0.745±0.015** | **-1.110±0.053** | **4.342±0.130** | **5.499±0.143** | **5.524±0.126** |
> | Syn-Test | C-FLUX             | **21.72±0.51** | 0.52±0.029 | 0.426±0.016 | 0.446±0.019 | **-1.085±0.223** | 0.688±0.020 | 72.349±1.41 | 0.748±0.022 | 0.717±0.022 | -1.158±0.077 | 4.443±0.121 | 5.374±0.162 | 5.325±0.141 |
> | Syn-Test | DP²O-SR (FLUX)     | 21.27±0.39 | 0.52±0.022 | **0.420±0.012** | **0.448±0.017** | -0.947±0.196 | **0.712±0.014** | **74.252±0.83** | **0.787±0.016** | **0.758±0.013** | **-1.210±0.065** | **4.577±0.083** | **5.462±0.136** | **5.443±0.117** |
> | RealSR   | C-SD2              | **23.38±0.79** | 0.62±0.036 | 0.467±0.019 | **0.441±0.019** | **-0.595±0.296** | 0.666±0.020 | 71.060±1.58 | 0.724±0.027 | 0.704±0.025 | -0.949±0.061 | 3.597±0.203 | 5.046±0.146 | 5.067±0.138 |
> | RealSR   | DP²O-SR (SD2)      | 22.73±0.62 | **0.64±0.026** | **0.440±0.012** | 0.436±0.016 | -0.356±0.183 | **0.700±0.014** | **72.710±0.96** | **0.758±0.021** | **0.747±0.015** | **-1.111±0.046** | **3.808±0.155** | **5.169±0.127** | **5.088±0.128** |
> | RealSR   | C-FLUX             | **24.18±0.81** | 0.66±0.037 | 0.410±0.021 | **0.481±0.026** | **-0.698±0.256** | 0.660±0.023 | 70.690±1.76 | 0.692±0.035 | 0.693±0.031 | -1.038±0.063 | 3.630±0.220 | 4.932±0.148 | 4.783±0.159 |
> | RealSR   | DP²O-SR (FLUX)     | 23.90±0.59 | 0.66±0.030 | **0.407±0.015** | 0.477±0.019 | -0.587±0.199 | **0.689±0.015** | **72.680±0.99** | **0.744±0.023** | **0.757±0.013** | **-1.088±0.051** | **3.903±0.143** | **5.021±0.119** | **4.948±0.120** |
>
> ---
>
> **[Q3]: Do CLIP-based CPS components favor oversharpened or hallucinated details?**
>
> Thank you for the insightful question.
>
> We have observed that CLIP-based components in CPS (e.g., CLIPIQA+, Q-Align) can indeed favor oversharpened or hallucinated textures when used in isolation.
>
> To address this, CPS integrates both FR and NR IQA metrics. The FR components serve to anchor the reward signal in ground-truth consistency, effectively mitigating the tendency toward hallucinated or overly sharp artifacts.
>
> We will include visual examples in the revised version to illustrate how CPS achieves a better balance between perceptual quality and fidelity.
>
> ---
>
> **[Q4]: Ablate HSW into intra-pair and inter-group to isolate contributions.**
>
> Thank you for the helpful suggestion.
>
> We have conducted separate ablation studies to isolate the effects of the two components in HSW:
>
> - (i) Intra-pair only (IP)
> - (ii) Inter-group only (IG)
>
> The table below shows the CPS score under different configurations:
>
> | Method            | CPS Score |
> |-------------------|-----------|
> | Baseline          | 0.588     |
> | + PPC             | 0.630     |
> | + PPC + IP only   | 0.634     |
> | + PPC + IG only   | 0.636     |
> | + PPC + IP + IG   | **0.642** |
>
> We find that both components individually contribute to performance gains, and their combination yields the best result.
>
> We will include this ablation in the revised manuscript to clarify the individual contributions of intra-pair and inter-group weighting.
>
> ---
>
> **[Q5]: Please provide justification of CPS composition and normalization.**
>
> Thank you for the helpful question.
>
> CPS is designed to balance **fidelity** and **perceptual quality** by combining both FR and NR IQA metrics:
>
> - **FR metrics** (e.g., LPIPS, TOPIQ-FR, AFINE-FR) emphasize structural and content fidelity.
> - **NR metrics** (e.g., CLIPIQA+, MUSIQ, Q-Align) encourage perceptual realism and aesthetics.
>
> Using only FR tends to suppress fine details, while NR-only optimization may lead to hallucinated or oversharpened artifacts. The combination helps avoid these extremes.
>
> To ensure fair contribution from each metric, we apply **min-max normalization** across each comparison set, aligning their numeric ranges. Additionally, we assign **equal weights to the FR and NR groups**, rather than individual metrics, to avoid imbalance due to differing group sizes.
>
> Metrics such as PSNR and SSIM are excluded from CPS due to their known poor correlation with human perception.
>
> Due to rebuttal format constraints, we are unable to include illustrative examples here, but we will provide detailed visual comparisons in the revised version to further justify the design of CPS.

---

> > ### Comment · Reviewer_Jm8D · 2025-08-04
> >
> > The rebuttal provides helpful clarifications, including a user study, single-sample inference results (M = 1), and ablation of the HSW loss. However, several core concerns remain unresolved, particularly around the evaluation design and generality of the proposed approach.
> >
> > 1. CPS metric serves as both the training signal and primary evaluation criterion, introducing a risk of circularity. While the authors reference a human preference study, it is reported only in the appendix without details on methodology, correlation with CPS, or how results generalize beyond the curated settings. Moreover, the CPS composition heavily depends on CLIP-based metrics (e.g., CLIPIQA+, Q-Align), which are known to favor oversharpened or hallucinated textures. These failure modes are acknowledged by the authors but not quantitatively analyzed or mitigated. Without a diagnostic study on hallucination sensitivity or metric manipulation, CPS remains a potentially brittle optimization target.
> >
> > 2.  Method is evaluated on only two diffusion backbones (SD2.0, FLUX) with a small training corpus, and performance is measured largely on internal datasets. There is no assessment on established perceptual SR benchmarks (e.g., NTIRE, PIPAL), nor is the generalization of DP²O-SR to other restoration architectures demonstrated. This limited scope restricts the impact and reproducibility of the method.
> >
> > 3. While the authors report mean ± std across random seeds, no statistical significance testing is performed. In some cases, performance improvements are within the reported variance, raising doubts about their robustness. Additionally, the CPS metric itself lacks a rigorous sensitivity or ablation analysis; the relative contributions of constituent metrics and the effect of group weighting strategies remain heuristic.
> >
> > While the connection between preference alignment and Real-ISR is timely and conceptually interesting, the current form of the paper suffers from evaluation circularity, metric brittleness, and limited empirical scope.

---

> > > ### Author Response · Authors · 2025-08-04
> > > **Response to Reviewer Jm8D's Comments**
> > >
> > > Thank you for the constructive feedback. We address the key concerns below.
> > >
> > > **Comment 1.1: Circularity and potential brittleness of CPS**
> > >
> > > Thank you for the insightful comment. CPS combines FR and NR metrics, most trained on human-labeled data and widely validated, enabling a balanced evaluation across diverse perceptual cues.
> > >
> > > To mitigate circularity, we report results on external metrics (e.g., NIMA, TOPIQ-IAA) not used in CPS. As shown in Table 1, our method consistently improves on these, suggesting perceptual gains generalize beyond the optimization signal.
> > >
> > > A user study and qualitative comparisons (Appendix Sec. 4–5) further confirm that DP²O-SR produces perceptually preferred results over baselines.
> > >
> > > **Comment 1.2: User study methodology and relation to CPS**
> > >
> > > Thank you for the suggestion. As described in Appendix Sec. 5, our user study involved 10 participants, 1,200 trials, and both pairwise and multi-way comparisons across two backbones.
> > >
> > > While we did not compute formal correlation coefficients, we observed strong alignment between CPS improvements and user preferences, particularly where DP²O-SR showed clear perceptual gains.
> > >
> > > We will highlight this alignment in the revision and plan formal correlation analysis in future work.
> > >
> > > **Comment 1.3: CPS vulnerable to CLIP hallucination**
> > >
> > > Thank you for raising this point. CLIP-based NR metrics (e.g., CLIPIQA+, Q-Align) may favor hallucinated details when used alone. CPS counteracts this by incorporating FR metrics that promote fidelity and balance potential biases.
> > >
> > > As shown in comparisons of FR-only, NR-only, and hybrid variants (see R`iowj` Q1), CPS yields more stable and perceptually aligned results. We will add qualitative examples in the revision to illustrate this trade-off.
> > >
> > > **Comment 2.1: Limited backbones and small training corpora**
> > >
> > > Thank you for the thoughtful suggestion. We selected two contrasting models—**SD2** (0.8B UNet, diffusion) and **FLUX** (12B DiT, flow-matching)—to span diverse scales, architectures, and training paradigms.
> > >
> > > This setup isolates the impact of the **CPS reward**, which is architecture-agnostic and independent of model-specific assumptions. We agree broader validation is valuable and plan to extend to more architectures and datasets in future work.
> > >
> > > **Comment 2.2: Evaluation on internal datasets; lack of public benchmarks**
> > >
> > > Thank you for the important suggestion. We agree public benchmarks are key to reproducibility.
> > >
> > > While our synthetic set is internal, we evaluate on **RealSR** and **DRealSR** (see R`Rx2M` W3), both widely adopted in perceptual SR. To further strengthen this, we also report results on the **PIPAL validation set**, a challenging public benchmark with 1000 LQ images from 25 references.
> > >
> > > Across a broad range of FR and NR metrics, **DP²O-SR** outperforms the baseline. Results are summarized below, and we hope this additional evaluation underscores the method’s robustness and generalization.
> > >
> > >
> > > | Method | LPIPS | TOPIQ-FR | AFINE-FR | MANIQA | MUSIQ | CLIPIQA | TOPIQ-NR | AFINE-NR | QALIGN | NIMA | TOPIQ-IAA |
> > > |-|-|-|-|-|-|-|-|-|-|-|-|
> > > | DP²O-SR (SD2) | 0.3366 | 0.5235 | -1.2836 | 0.7349 | 70.93 | 0.7573 | 0.7583 | -1.3060 | 3.2125 | 5.06 | 4.1780 |
> > > | C-SD2         | 0.3432 | 0.5225 | -0.7594 | 0.6719 | 66.95 | 0.7187 | 0.7251 | -1.0885 | 3.0275 | 4.96 | 4.1723 |
> > >
> > > **Comment 3.1: Statistical significance testing**
> > >
> > > Thank you for the suggestion. We performed **paired t-tests** on both FR and NR metrics at the 5% significance level.
> > >
> > > All improvements are statistically significant (p < 0.05), confirming the gains are consistent and unlikely due to chance.
> > >
> > > Notably, DP²O-SR improves both FR and NR metrics—a rare outcome in perceptual SR that underscores its balanced effectiveness.
> > >
> > > | IQA | Before | After | p-value | Significant |
> > > |-|-|-|-|-|
> > > | LPIPS | 0.4577 | 0.4466 | <0.001 | ✅
> > > | TOPIQ-FR | 0.4128 | 0.4184 | 0.021 | ✅
> > > | AFINE-FR | -0.8116 | -1.0084 | 0.004 | ✅
> > > | CLIPIQA | 0.7263 | 0.7755 | <0.001 | ✅
> > > | MANIQA | 0.6564 | 0.6865 | <0.001 | ✅
> > > | MUSIQ | 70.70 | 73.77 | <0.001 | ✅
> > > | TOPIQ-NR | 0.6946  | 0.7446 | <0.001 | ✅
> > > | AFINE-NR | -0.924  | -1.110 | <0.001 | ✅
> > > | Q-Align | 4.050 | 4.342 | <0.001 | ✅
> > >
> > > **Comment 3.2: Sensitivity and weighting strategies in CPS and HSW**
> > >
> > > Thank you for the thoughtful feedback. We agree that CPS and HSW should be empirically grounded and transparent.
> > >
> > > For CPS, our sensitivity analysis (see R`iowj` Q2) shows that while metrics like AFINE-FR and MUSIQ contribute more, no single metric dominates. Ablations using top-1/2/3 metrics led to consistent drops, confirming the complementary roles of FR and NR metrics. Equal intra-group weighting (FR: 1/6, NR: 1/12) also outperformed reweighted or reduced variants.
> > >
> > > For HSW, the group-aware design promotes both contrast and diversity, yielding richer supervision than binary wins/losses and improving stability across settings.
> > >
> > > We will clarify these findings to better support the empirical basis of both strategies.

---

> > > > ### Comment · Reviewer_Jm8D · 2025-08-05
> > > >
> > > > The rebuttal addresses many of the key concerns. The authors add public benchmark results (PIPAL), report statistical significance for gains, and provide details of a user study showing perceptual preference alignment. These significantly strengthen the empirical validity of the approach.
> > > >
> > > > Some limitations remain: the method is evaluated on only two diffusion-based models, and the CPS metric's robustness to CLIP-induced hallucination is not fully quantified. Still, the method represents a promising adaptation of preference learning to SR, and the presented results are compelling.
> > > >
> > > > I will revise my score acknowledging the paper’s technical merit and growing relevance of preference-driven optimization in perceptual image restoration.

---

> > > > > ### Author Response · Authors · 2025-08-05
> > > > > **Response to Reviewer Jm8D's Comment**
> > > > >
> > > > > Thank you for your thoughtful follow-up and for acknowledging the improvements and contributions. We truly appreciate your willingness to revise the score, and your feedback on remaining limitations will be valuable for future iterations of this work.

---

> ### Author Response · Authors · 2025-08-04
>
> Dear Reviewer Jm8D,
>
> Many thanks for your time in reviewing our paper and your constructive comments. We have submitted the point-to-point responses. We appreciate if you could let us know whether your concerns have been addressed, and we are happy to answer any further questions.
>
> Best regards,
>
> Authors of paper \#5066

---

### Official Review · Reviewer_iowj · 2025-07-02

**Clarity:** 3
**Significance:** 2
**Originality:** 3
**Rating:** 4
**Confidence:** 3

**Summary:**

The authors propose a novel model for Real-World Image Super-Resolution. The approach utilizes a pretrained model as a reference for generating super-resolution (SR) candidates from a given low-resolution input. These candidates are then evaluated using a composite metric called CPS, which ranks them from top to bottom. In the subsequent training phase, one top-ranked and one bottom-ranked candidate are selected and combined for optimization using Direct Preference Optimization (DPO).

**Questions:**

* What is the impact of using only FR or NR metrics in the model?

* Which IQA metric is most similar to CPS?. This analysis could provide deeper insight into why CPS performs well for both ranking and significance weighting.

* Why are FR and NR metrics given equal importance, considering NR includes six metrics while FR reports only three?

* What is the performance gap when using random pair selection instead of hierarchical significance weighting?. This experiment can help establish a lower bound for comparison with the proposed approach.

* Can the intra-pair strategy be implemented by simply selecting the top- or bottom-1 candidate and selecting all counterparts' samples (i.e., selecting top-1 and then selecting several samples of the bottom part? Or is an additional condition required to maximize variance?

* Is there any negative impact if the dataset contains an imbalance between positive and negative samples?

**Ethical Concerns:**

["NO or VERY MINOR ethics concerns only"]

**Final Justification:**

The authors did an excellent job clearing my concerns. I believe that the paper will be improved with the new details and results.  I raised my vote to weak accept

**Limitations:**

yes

**Paper Formatting Concerns:**

there is no concerns

**Quality:**

2

**Strengths And Weaknesses:**

While the model demonstrates superior performance compared to all baselines, several concerns remain regarding the clarity and justification of key components. First, CPS is introduced as a composite metric, however, there is no clear motivation for using all IQA metrics. A more principled justification or ablation study would strengthen the argument. Second, lines 135–136 state that “CPS is a relative score computed within the compared set S,” yet Tables 1, 2, and 3 report CPS scores across different models. This contradicts the definition and raises questions about the validity and interpretability of these comparisons between models. Third (minor), the intended message of Figure 3a-b is unclear. Both images appear visually similar, and the role of the downward arrow is not explained, reducing the figure’s impact. Fourth, Section 3.2 describes a core component of the model, but is only briefly described in the main paper. Key algorithmic steps and equations are presented in the supplementary material, which may hinder comprehension for readers.

---

> ### Author Rebuttal · Authors · 2025-07-30
>
> We thank the reviewer iowj  for his/her valuable feedback. For brevity, we use `W#` for weaknesses and `Q#` for questions (e.g., W1, Q1) throughout our response.
>
> ---
>
> **[W1, Q1 and Q3]: Why use all IQA metrics? Why not only use FR or NR metrics as reward? Why are FR and NR metrics given equal importance in CPS formulation?**
>
> We thank the reviewer for the insightful questions.
>
> We do not use all IQA metrics indiscriminately. Instead, we carefully select 3 full-reference (FR) and 6 no-reference (NR) metrics commonly used in perceptual quality assessment.
>
> FR metrics assess fidelity and structural similarity to ground truth, but relying solely on them can produce less vivid outputs. NR metrics promote perceptual quality without reference, encouraging more appealing visuals. However, NR metrics alone may lead to reward hacking or favor unrealistic generations—a concern also raised by Reviewer Jm8D (Q2) regarding CLIP-based NR metrics.
>
> To investigate further, we conducted ablation studies using only FR or only NR metrics in CPS. As shown below, FR-only rewards improve fidelity but hurt perceptual quality, while NR-only rewards do the opposite. Combining both gives a better overall balance:
>
> | Method        | LPIPS ↓ | TOPIQ-FR ↑ | AFINE-FR ↓ | MANIQA ↑ | MUSIQ ↑ | CLIPIQA+ ↑ | TOPIQ-NR ↑ | AFINE-NR ↓ | QALIGN ↑ |
> |---------------|---------|-------------|-------------|-----------|----------|--------------|--------------|--------------|------------|
> |               |   FR    |     FR      |     FR      |    NR     |   NR     |     NR       |     NR       |     NR       |    NR      |
> | Only FR       | 0.428   | 0.442       | -1.144      | 0.631     | 70.5     | 0.724        | 0.689        | -1.003       | 4.083      |
> | Only NR       | 0.476   | 0.389       | -0.745      | 0.722     | 74.6     | 0.795        | 0.762        | -1.142       | 4.289      |
> | FR + NR (ours)| 0.447   | 0.418       | -1.008      | 0.687     | 73.8     | 0.776        | 0.745        | -1.110       | 4.342      |
>
> To avoid imbalance from the unequal number of FR and NR metrics, we assign equal total weight to each group in the CPS formulation. This prevents overemphasizing the NR group and ensures that both fidelity and perceptual quality are equally represented during optimization.
>
> We agree that visual examples would help illustrate these trade-offs. Due to rebuttal constraints, we will include qualitative comparisons in the final version.
>
> ---
>
> **[W2]: Why are CPS scores compared across models if CPS is defined as relative within a set?**
>
> Thank you for the thoughtful comment and for highlighting this potential confusion.
>
> To clarify, CPS is not computed by normalizing each model’s outputs independently. Instead, for each input image, we collect outputs from *all models under comparison* (e.g., those in the same table), and apply min-max normalization jointly across this full set.
>
> This ensures CPS reflects the relative perceptual quality across models, making scores directly comparable within the same evaluation group. Thus, CPS values in Tables 1–3 are valid for intra-table comparison, but not across different tables or experiments.
>
> We appreciate the reviewer’s attention and will clarify this computation procedure in the revised manuscript.
>
> ---
>
> **[W3]: HSW lacks detail in the main paper, relying on the supplement.**
>
> Thank you for the suggestion. We will add the key HSW steps and equations to the main paper to improve clarity and self-containment.
>
> ---
>
> **[W4]: The purpose of Figure 3a-b is unclear, and the downward arrow is unexplained.**
>
> Thank you for the feedback. We will revise Figure 3 to clarify its message: (a) shows that increasing sampling times during preference data construction improves performance with best/worst-of-1 pairs; (b) illustrates a trade-off between data quality and quantity when using best/worst-of-N, with optimal performance at a moderate N.
>
> The downward arrow indicates increased sampling during the preference data collection stage.
>
> ---
>
> ### **[Q2]: Which IQA metric aligns most closely with CPS?**
>
> Thank you for the thoughtful question. We analyzed which IQA metrics align most closely with CPS and whether a reduced metric set could suffice for reward design.
>
> CPS is computed as the average of normalized FR and NR metrics. To ensure equal group contributions, each FR metric is weighted 1/6 and each NR metric 1/12. The table below shows the normalized scores of our model (DP²O-SD2), the weights, and each metric’s contribution:
>
> | Metric     | Group | Normalized Score | Weight | Contribution |
> |------------|--------|------------------|--------|--------------|
> | LPIPS      | FR     | 0.5015           | 1/6    | 0.0836       |
> | TOPIQ-FR   | FR     | 0.4777           | 1/6    | 0.0796       |
> | AFINE-FR   | FR     | 0.7072           | 1/6    | 0.1179       |
> | MANIQA     | NR     | 0.6693           | 1/12   | 0.0558       |
> | MUSIQ      | NR     | 0.8318           | 1/12   | 0.0693       |
> | CLIP-IQA   | NR     | 0.7228           | 1/12   | 0.0602       |
> | TOPIQ-NR   | NR     | 0.8167           | 1/12   | 0.0681       |
> | AFINE-NR   | NR     | 0.5725           | 1/12   | 0.0477       |
> | QALIGN     | NR     | 0.7632           | 1/12   | 0.0636       |
> | **Total**  | —      | —                | 1      | 0.6458       |
>
> Among FR metrics, AFINE-FR, LPIPS, and TOPIQ-FR contribute most to CPS. For NR metrics, MUSIQ, TOPIQ-NR, and QALIGN are the top contributors.
>
> To test if a reduced set suffices, we conducted ablations using only the top-k metrics from each group. This reflects the RealSR task’s need to balance **fidelity** (FR) and **perception** (NR). Results are shown below:
>
> | Metric     | Top-1 FR+NR (AFINE-FR, MUSIQ) | Top-2 FR+NR (+LPIPS, TOPIQ-NR) | Top-3 FR+NR (+TOPIQ-FR, QALIGN) | All metrics (ours) |
> |------------|------------------|-------------------|--------------------|------------------|
> | LPIPS      | 0.4694           | 0.4513            | 0.4461             | 0.4466           |
> | TOPIQ-FR   | 0.3974           | 0.4122            | 0.4216             | 0.4184           |
> | AFINE-FR   | -0.9426          | -0.9759           | -1.0110            | -1.0084          |
> | MANIQA     | 0.7011           | 0.6890            | 0.6806             | 0.6865           |
> | MUSIQ      | 74.2999          | 73.7008           | 73.5275            | 73.7686          |
> | CLIP-IQA   | 0.7719           | 0.7761            | 0.7755             | 0.7755           |
> | TOPIQ-NR   | 0.7418           | 0.7509            | 0.7466             | 0.7446           |
> | AFINE-NR   | -1.0903          | -1.0808           | -1.0927            | -1.1095          |
> | QALIGN     | 4.2324           | 4.2779            | 4.3241             | 4.3424           |
> | CPS↑       | 0.6223           | 0.6330            | 0.6536             | 0.6909           |
>
> While CPS is most influenced by a few strong metrics, using only top-k metrics consistently underperforms compared to using all. This suggests that the additional metrics provide **complementary, non-redundant signals**, supporting the need for a diverse set in reward design.
>
> We will include this analysis and tables in the revised manuscript.
>
> ---
>
> **[Q4]: What's the performance drop when replacing HSW with random pairing?**
>
> Thank you for the helpful question.
>
> Our HSW strategy builds on the PPC sampling framework:
>
> - The **baseline** uses a fixed top-1 vs. bottom-1 pairing.
> - **PPC** introduces diversity by randomly pairing samples within top-N and bottom-N candidates.
> - **HSW** further improves this by assigning higher weights to more informative pairs during training.
>
> As shown in Table 2(c), PPC improves CPS from 0.626 (baseline) to 0.632, and adding HSW further raises it to 0.641.
>
> We do not perform fully random pairing (i.e., sampling any two outputs arbitrarily), as our method relies on reward-ranked candidates to ensure meaningful preference signals.
>
> We will clarify this distinction and the ablation structure in the revised manuscript.
>
> ---
>
> **[Q5 & Q6]: How do intra-pair strategies and class imbalance affect model performance? Is a simple top-1 vs. bottom-N pairing sufficient?**
>
> Thank you for the insightful questions.
>
> **For Q5:**
> We tested a top-1 vs. bottom-N pairing strategy, where the best sample is paired with multiple negatives. As shown in the table (e.g., *Top1+Bottom10*, CPS = 0.685), this setup performs reasonably well but consistently lags behind our full method (CPS = 0.703).
>
> This suggests that fixing the anchor to top-1 limits intra-pair diversity. In contrast, sampling from top-N and bottom-N introduces richer variation, which benefits preference learning.
>
> **For Q6:**
> We agree that imbalance can bias training. To mitigate this, we maintain symmetry between positive and negative pools (top-N vs. bottom-N). While exhaustive testing is infeasible, our experiments with asymmetric settings (e.g., Top1+Bottom4/7/10) show diminishing gains, supporting the effectiveness of balanced sampling.
>
> Overall, our strategy balances informativeness and robustness, which we find crucial for stable optimization. We will clarify these points in the revised version.
>
> | Method           | LPIPS↓ | TOPIQ-FR↑ | AFINE-FR↓ | MANIQA↑ | MUSIQ↑ | CLIPIQA↑ | TOPIQ-NR↑ | AFINE-NR↓ | QALIGN↑ | CPS↑  |
> |------------------|--------|-----------|-----------|---------|--------|----------|-----------|------------|---------|--------|
> | Top1+Bottom4     | 0.460  | 0.403     | -0.909    | 0.686   | 74.12  | 0.765    | 0.750     | -1.140     | 4.239   | 0.669  |
> | Top1+Bottom7     | 0.454  | 0.411     | -0.896    | 0.684   | 74.11  | 0.773    | 0.753     | -1.152     | 4.224   | 0.689  |
> | Top1+Bottom10    | 0.453  | 0.409     | -0.874    | 0.681   | 74.15  | 0.769    | 0.755     | -1.145     | 4.256   | 0.685  |
> | Ours             | 0.447  | 0.418     | -1.008    | 0.687   | 73.80  | 0.776    | 0.745     | -1.110     | 4.342   | 0.703  |

---

> > ### Comment · Reviewer_iowj · 2025-08-04
> >
> > Thank you to the authors for providing a thorough explanation. My concerns have been completely resolved, and I will adjust my vote accordingly.

---

> > > ### Author Response · Authors · 2025-08-05
> > > **Response to Reviewer iowj's Comments**
> > >
> > > Thank you for your kind and thoughtful reconsideration. We’re grateful our response helped, and we truly appreciate your support.

---

> ### Author Response · Authors · 2025-08-04
>
> Dear Reviewer iowj,
>
> Many thanks for your time in reviewing our paper and your constructive comments. We have submitted the point-to-point responses. We appreciate if you could let us know whether your concerns have been addressed, and we are happy to answer any further questions.
>
> Best regards,
>
> Authors of paper \#5066

---

### Comment · Area_Chair_9h2i · 2025-08-04
**Please discuss with the authors**

Dear Reviewers,

Please discuss with the authors, especially if the rebuttal did not solve your concerns.

Best,
Your AC

---

### Author Response · Authors · 2025-08-09
**Response Summary to Reviewers and Area Chairs**

Dear Reviewers and Area Chairs,

We sincerely thank all reviewers for their valuable feedback and active engagement during the discussion phase. We are encouraged that all reviewers have expressed a willingness to maintain or revise toward a positive rating, and we greatly appreciate the constructive spirit throughout the exchange.

Specifically:
- **R-iowj** confirmed all concerns were resolved and increased the score.
- **R-Jm8D** acknowledged the added experiments and stated intent to revise the score, recognizing the paper’s technical merit and relevance.
- **R-Rx2M** consistently provided a high-confidence, positive review and helpful suggestions.
- **R-4d1m** appreciated the improvements and moved to a more positive score.

We have responded to each reviewer in detail and would like to summarize the main clarifications and additional experiments below:

---

### **Additional Experiments**
- **Single-sample Evaluation**: Reported CPS, LPIPS, PSNR, SSIM under M = 1 to validate practical use.
- **Statistical Significance**: Added t-tests (p < 0.05) and variance across 20 random seeds.
- **Public Benchmarks**: Newly evaluated on **PIPAL** and **DRealSR** to demonstrate generalization.
- **HSW Ablation**: Isolated intra-pair and inter-group effects to validate the weighting scheme.
- **Candidate Pairing Strategies**: Compared Top1-vs-BottomN pairing with full PPC+HSW design.
- **Metric Sensitivity**: Analyzed CPS under different metric subsets to support full composition.

---

### **Clarifications & Explanations**
- **User Study**: Already reported in Appendix Sec. 5 (1,200 trials); now emphasized as human validation of CPS.
- **CPS Normalization**: Clarified joint normalization across models and equal FR/NR weighting.
- **CLIP-induced Hallucination**: Explained how FR metrics in CPS constrain CLIP-related artifacts.
- **Candidate Generation Cost**: Reported runtime on 8×A100 GPUs and compared offline vs. online RL.
- **Sampling Strategy**: Justified fixing CFG and varying noise seeds to ensure controlled diversity.

---

We hope these efforts demonstrate the rigor and potential impact of our work. Thank you again for your time and thoughtful consideration.

Best regards,

*Authors of paper #5066*

---

### Decision · Program_Chairs · 2025-09-17

**Decision:**

Accept (poster)

**Comment:**

This submission presents a method called DP²O-SR for real-world image super-resolution that leverages Direct Preference Optimization to align diffusion model outputs with human perceptual preferences. The key contributions include the Combined Perceptual Score (CPS) for quantifying perceptual quality and Hierarchical Significance Weighting (HSW) to enhance training efficiency. Reviewers acknowledged the submission's motivation, methodological innovation, and empirical improvements over baselines, particularly in perceptual quality and stability. While initial concerns were raised about metric circularity (CPS serving as both training signal and evaluation criterion), the authors addressed these through additional experiments. The rebuttal further clarified computational costs, failure modes, and the balance between fidelity and perceptual quality. Although some limitations remain, the reviewers unanimously upgraded their scores, recognizing the submission's technical rigor, practical impact. The submission is thus suggested for acceptance.